# Mitophagy curtails cytosolic mtDNA-dependent activation of cGAS/STING inflammation during aging

Juan Ignacio Jiménez-Loygorri [1], Beatriz Villarejo-Zori[1], Álvaro Viedma-Poyatos[1], Juan Zapata-Muñoz[1], Rocío Benítez-Fernández [1,2], María Dolores Frutos-Lisón[3], Francisco A. Tomás-Barberán [3], Juan Carlos Espín [3], Estela Area-Gómez[4], Aurora Gomez-Duran [4,5] & Patricia Boya [1,2] ✉

Macroautophagy decreases with age, and this change is considered a hallmark of the aging process. It remains unknown whether mitophagy, the essential selective autophagic degradation of mitochondria, also decreases with age. In our analysis of mitophagy in multiple organs in the mito-QC reporter mouse, mitophagy is either increased or unchanged in old versus young mice. Transcriptomic analysis shows marked upregulation of the type I interferon response in the retina of old mice, which correlates with increased levels of cytosolic mtDNA and activation of the cGAS/STING pathway. Crucially, these same alterations are replicated in primary human fibroblasts from elderly donors. In old mice, pharmacological induction of mitophagy with urolithin A attenuates cGAS/STING activation and ameliorates deterioration of neurological function. These findings point to mitophagy induction as a strategy to decrease age-associated inflammation and increase healthspan.

Autophagy is one of the main intracellular recycling mechanisms responsible for maintaining homeostasis[1]. Mitophagy is defined as the removal of faulty or superfluous mitochondria by autophagy machinery, and can be mediated by the PTEN-induced kinase 1 (PINK1)/Parkin pathway, mitophagy receptors, or changes in mitochondria lipid composition[2,3]. Intracellular housekeeping is particularly important in the context of aging, as post-mitotic tissues must manage the accumulation of cellular debris caused by a lifetime of internal and external stressors[1]. Deficient autophagy has been recently proposed as a hallmark of aging[4,5]. In the retina, alternatively, more selective pathways such as chaperone-mediated autophagy can compensate for the effects of age-associated decreases in autophagy[6]. While some studies have reported age-associated downregulation of mitophagy[7,8], organ-specific increases in mitophagy have also been described[9–12]. We conducted a systematic and systemic analysis of age-associated changes in mitophagy in mice, examined the interplay between mitophagy and mtDNA-triggered inflammation, and explored the potential of mitophagy-boosting strategies to alleviate age-associated physiological decline.

## Results

### Mitophagy levels increase or remain stable during physiological aging

Changes in mitophagy associated with aging have been studied extensively[7–12]. However, whether mitophagy is up- or downregulated in old tissues remains unclear. This is partly due to the scarcity and complexity of currently available tools and read outs for the study of mitophagy[13]. In the present study, we assessed mitophagy using the fixable *mito*-QC reporter C57BL/6J mouse[14], which ubiquitously expresses a chimeric protein that consists of pH-insensitive mCherry,

[1]Department of Cellular and Molecular Biology, Centro de Investigaciones Biológicas Margarita Salas, CSIC, Madrid, Spain. [2]Department of Neuroscience and Movement Science, Section of Medicine, University of Fribourg, Fribourg, Switzerland. [3]Food & Health Lab, Research Group on Quality, Safety, and Bioactivity of Plant Foods, CEBAS-CSIC, Murcia, Spain. [4]Department of Biomedicine, Centro de Investigaciones Biológicas Margarita Salas, CSIC, Madrid, Spain. [5]MitoPhenomics Lab, Centro Singular de Investigación en Medicina Molecular y Enfermedades Crónicas, Universidade de Santiago de Compostela, Santiago de Compostela, Spain. ✉ e-mail: patricia.boya@unifr.ch

pH-sensitive GFP, and the mitochondria-targeting sequence (MTS) of mitochondrial fission 1 protein (FIS1) (Fig. 1a). Thus, microscopy can be used to distinguish cytoplasmic mitochondria (mCherry⁺GFP⁺) from those undergoing lysosomal degradation (mCherry⁺GFP⁻). Confocal imaging analysis of tissue samples from young (6–8 months) and old (24–26 months) mice revealed significantly higher levels of mitophagy in the kidney, brain, retinal pigment epithelium (RPE), neuroretina, cerebellum, and liver in old mice (Fig. 1b, Supplementary Fig. 1a–e). By contrast, mitophagy levels in the pancreas, spleen, muscle, heart, and lungs did not differ between groups (Fig. 1b). Previous reports in the literature have shown that deficient lysosomal degradation can lead to confounding interpretation of experiments using tandem fluorescent reporters[15]. Since deficient lysosomal function has been associated with aging[16], we evaluated mitolysosome levels in the retina of young and old mice treated with the protease inhibitor Leupeptin (40 mg/kg, 16 h) or vehicle (saline). Supporting our initial findings (Fig. 1b), we indeed observed a significant and more acute accumulation of mitolysosomes, or mitophagic flux, in the old group during the 16-h treatment time window (Supplementary Fig. 1f).

Increased PINK1-induced phosphorylation of Ubiquitin at Ser65 was observed in most organs analyzed (Fig. 1c, d, Supplementary Fig. 2a), indicating that the age-associated increase in mitophagy is driven by the PINK1/Parkin pathway that responds to perturbations in mitochondrial membrane potential[17]. We also measured the levels of proteins involved in receptor-mediated mitophagy (BNIP3L/NIX, BNIP3, FKBP8, PHB2, FUNDC1) in the retina and found no age-related changes (Supplementary Fig. 2b). Similarly, even though the levels of cardiolipin were significantly decreased in the retina of old mice, there were no changes in cardiolipin translocation to the outer mitochondrial membrane (OMM; Supplementary Fig. 2c) discarding the involvement of lipid-mediated mitophagy.

In line with previous reports from our group[6] and others[4], old mice displayed a reduction in general macroautophagy compared to young mice, as evidenced by an increase in autophagosome number (LC3⁺) that was not accompanied by a change in autolysosome number (LC3⁺LAMP1⁺) (Supplementary Fig. 3a, b). These observations in old mice, together with the accumulation of the autophagy adaptor Sequestosome-1/p62 and poly-ubiquitinated proteins (Supplementary Fig. 3c, d), as well as a lack of transcriptional changes in autophagy regulators (Supplementary Fig. 3e, f), point towards a generalized blockade of macroautophagic flux that coincides with an increase in mitophagy. Because mitochondrial dysfunction has also been proposed as one of the main drivers of aging[4], we sought to identify the causes and consequences of the shift accompanying physiological aging, whereby degradation of mitochondria is favored over other autophagic substrates.

## Free cytosolic mtDNA triggers a cGAS/STING-mediated type I interferon response and inflammation in old mice

The retina is a well-defined, easily accessible part of the central nervous system that displays one of the most robust age-associated increases in mitophagy of all organs. Transcriptomic analysis of retinas from young and old mice (Supplementary Fig. 4a) revealed that several of the most downregulated signaling pathways in old mice were associated with mitochondria (Supplementary Fig. 4b, c; detailed list included in Source Data). However, these changes did not translate into decreases in structural proteins (Translocase of Inner Mitochondrial Membrane 23 (TIMM23); Supplementary Fig. 4d), mitochondrial mass (as a surrogate Succinate Dehydrogenase subunit B (SDHB)/CII) or the main components of OxPhos complexes (NADH:Ubiquinone Oxidoreductase Core Subunit S1 (NDUFS1)/CI, Ubiquinol-Cytochrome C Reductase Core Protein 2 (UQCRC2)/CIII, Mitochondrially Encoded Cytochrome C Oxidase I (MT-CO1)/CIV, ATP synthase subunit alpha (ATP5A)/CV; Supplementary Fig 4d). Except for a slight increase in brain and muscle in the old group (Supplementary Fig. 4e), most

organs showed similar levels of mitochondrial mass, indicating that overall mitochondria levels remain stable with increasing age. Nonetheless, in old mice electron microscopy revealed an increase in the number of swollen mitochondria and disruption of both inner and outer mitochondrial membranes (Supplementary Fig. 4f), a phenomenon linked to the release of its content including cytochrome C and mtDNA[18,19].

Retinas from older mice showed upregulation of several pathways implicated in inflammation and the response to viral or endogenous cytosolic DNA (Fig. 2a). Indeed, the top three positively enriched hallmarks were interferon response (INFα, IFNγ) and TNFα signaling (Fig. 2b). A more refined analysis using Interferome[20] showed that 67.5% of the significantly upregulated, differentially-expressed genes (DEGs) corresponded to interferon-stimulated genes (ISGs), and all were involved in the type I interferon response (Fig. 2c). Given the immune-privileged nature of the retina[21] and the strict veterinary monitoring that all mice in this study underwent, we hypothesized that endogenous cytosolic DNA might trigger this inflammatory response. Anti-DNA immunostaining of young and old retinas revealed an increase in the amount of cytosolic DNA in the old group, most of which were located in mitochondria-rich regions (Fig. 2d). Subcellular fractionation of retina samples was used to further explore the increase in cytosolic DNA in old mice, revealing a significant, 10-fold increase in intact mtDNA levels (mt-Nd2, mt-Co1, mt-Cytb) in the cytosolic fraction in old versus young mice (Fig. 2e).

The cyclic GMP-AMP synthase–stimulator of interferon gene (cGAS/STING) axis has been identified as one of the main sensors and downstream effectors of mtDNA release-triggered inflammation[22]. In keeping with the transcript data, we observed a robust increase in protein levels of both cGAS and STING (Fig. 2f), leading to activation of the downstream transcription factor Interferon Regulatory Factor 3 (IRF3), as reflected by increased levels of phospho-IRF3^Ser396 (Fig. 2f). Transcript levels of several genes downstream of IRF3 (Ifit1, Ifi44, Ifih1, Cxcl10, Rtp4) were also significantly upregulated in old retinas (Fig. 2g). Because the age-associated increase in mitophagy was observed across multiple organs (Fig. 1b), we also assessed cGAS/STING activation in the liver, brain, kidney, and muscle (Supplementary Fig. 5a). STING levels were significantly higher in kidney and muscle samples from old versus young mice (Supplementary Fig. 5b). To corroborate these findings, we analyzed levels of cGAS/STING mediators and downstream target genes across 17 different organs and five distinct age groups (4, 9, 12, 18, 24 months) using public transcriptomic data (GSE141252)[23] from C57BL/6J mice (Supplementary Fig. 5c). A common pattern towards increased transcript levels was observed in 12 (adrenal gland, brown adipose tissue, cerebellum, brain frontal cortex, heart, kidney, large intestine, liver, lung, muscle, skin, and white adipose tissue) out of 17 organs analyzed (Supplementary Fig. 5d), indicating that the increase in cGAS/STING-mediated inflammation in old mice is not limited exclusively to the retina.

Finally, we investigated whether the increase in mitophagy, mtDNA release, and cGAS/STING activation observed in old mice are also conserved across species. First, we performed a transcriptomic analysis of previously published data (GSE113957)[24] from primary normal human dermal fibroblasts (NHDF) that were categorized as Pediatric, Young Adult, Adult, Middle-aged Adult, Elderly, or Geriatric following clinical guidelines (Fig. 3a)[25]. Mirroring our findings in mice, the Geriatric group showed the highest levels of cGAS/STING mediators and downstream IRF3 targets (Fig. 3b), and enrichment in interferon response (INFα, IFNγ) and TNFα signaling with respect to all other groups (Fig. 3b). Furthermore, of the DEGs that were upregulated in the Geriatric group compared to all others, 50.3% corresponded to ISGs and were classified as interferon type I response genes (Fig. 3d). A correlation analysis of IFIH1 and RTP4 (downstream targets of IRF3) expression revealed a positive correlation with age (Fig. 3e). Finally, in-house analysis of NHDFs from young (28 years) and old (62

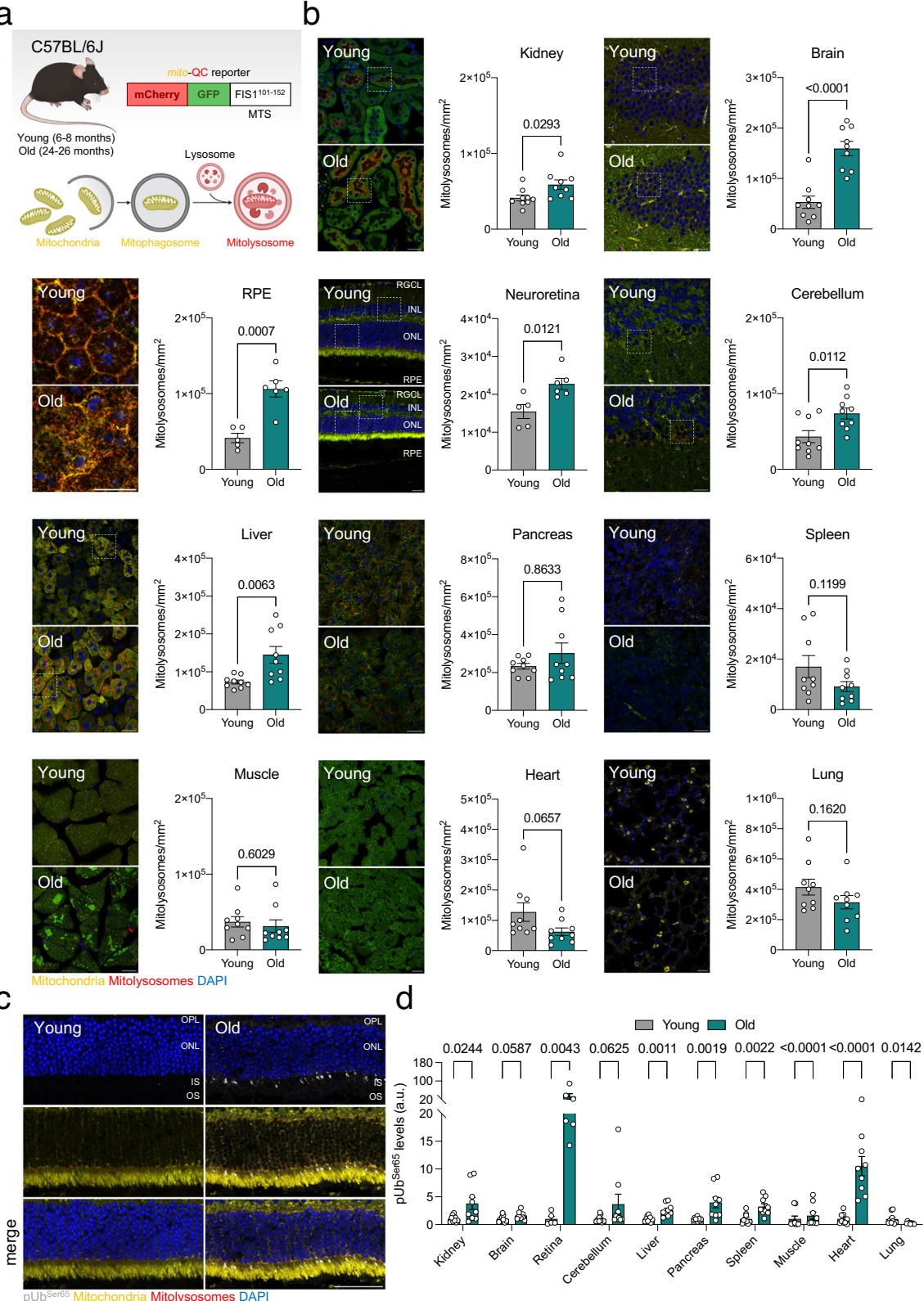

**Fig. 1 | Physiological aging in mice is associated with stable or increased mitophagy in multiple organs. a** Young (6–8 months) and old (24–26 months) *mito*-QC reporter mice bred on a C57BL/6J background were sacrificed and processed for confocal analysis. Created with BioRender.com. **b** Representative images and quantification of mitolysosome number (mCherry$^+$GFP$^-$ puncta) in the kidney (renal cortex), brain (hippocampus), RPE, neuroretina, cerebellum, liver, pancreas (exocrine), spleen, muscle (gastrocnemius), heart, and lungs (n = 5–9 mice). Higher magnification insets are provided in Supplementary Fig. 1. **c** Representative images of whole eye cryosections from young and old *mito*-QC mice

immunostained for phospho-Ubiquitin$^{Ser65}$ (gray). **d** Quantification of phospho-Ubiquitin$^{Ser65+}$ area in organs from young and old *mito*-QC mice (n = 5–9 mice). Scale bars, 25 μm (**b**) and 50 μm (**c**). All data are expressed as the mean ± s.e.m. Dots represent individual mice. P values were calculated using a two-tailed Student's *t* test (**b**, Kidney, brain, RPE, neuroretina, cerebellum, liver, spleen, muscle, heart. **d**. Brain, liver, spleen, muscle) or two-tailed Mann–Whitney *U*-test (**b** Pancreas. **d** Kidney, neuroretina, cerebellum, pancreas, heart, lung). Source data are provided as a Source Data file.

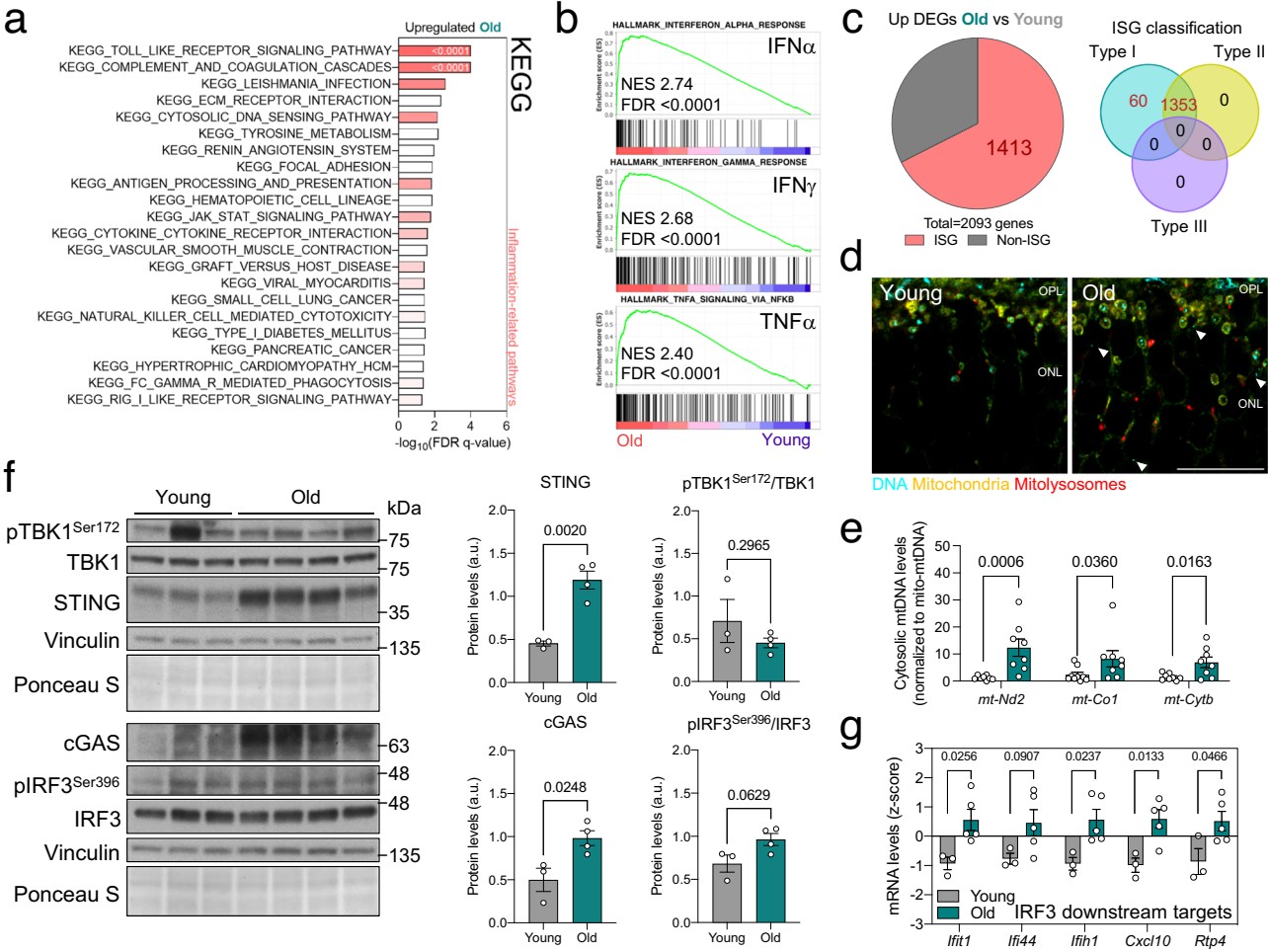

**Fig. 2 | Increased mtDNA release in old mice triggers cGAS/STING-mediated inflammation.** **a** Bulk RNA-seq data showing the top enriched KEGG pathways in old versus young retinas. Most correspond to inflammation-related signaling (color intensity corresponds to FDR q-value). **b** Top three enriched Hallmarks in old retinas: IFNα and IFNγ responses and TNFα signaling. **c** Interferome analysis (left) of upregulated DEGs in old retinas, showing the proportion of interferon-stimulated genes (ISG, 67.51%) and corresponding ISG classification (right). **d** Anti-DNA (cyan) immunostaining of whole eye cryosections from young and old *mito*-QC mice. Arrowheads indicate cytoplasmic DNA (GFP⁻DNA⁺) and the outer nuclear layer is shown. **e** qPCR measurement of retinal cytosolic mtDNA in young and old mice (*mt-Nd2, mt-Co1, mt-Cytb*), normalized to mtDNA content in the mitochondrial fraction ($n = 8–9$ mice). **f** Western blot analysis of cGAS/STING mediators (TBK1, IRF3, cGAS, STING) in young and old retinas ($n = 3–4$ mice). Protein levels are normalized to the loading control (Vinculin). **g** mRNA levels of downstream targets of the transcription factor IRF3 obtained from retina bulk RNA-seq data ($n = 3–5$ mice). Scale bar, 15 μm (**d**). All data are expressed as the mean ± s.e.m. Dots represent individual mice. *P* values were calculated using a two-tailed Student's *t* test (**e** (*mt-Nd2*), **f**, **g**) or two-tailed Mann–Whitney *U*-test (**e** (*mt-Co1, mt-Cytb*). Source data are provided as a Source Data file.

years) donors (Fig. 3f, g) showed significantly higher levels of phospho-Ubiquitin^Ser65 (Fig. 3h) and cytosolic DNA foci (Fig. 3i) in the latter group. Similar to the phenotype observed in the retina of old mice (Supplementary Fig. 2b, c), no changes were observed in the levels of receptor-mediated mitophagy effectors (Supplementary Fig. 6a) nor on cardiolipin translocation to the OMM (Supplementary Fig. 6b). These data support an age-dependent association between increases in cGAS/STING-mediated inflammation and mitophagy, and suggest that this association is conserved across organs and species. These observations suggest that PINK1/Parkin-mediated mitophagy may be selectively upregulated as aging progresses to improve mitochondrial quality control and counteract mtDNA release, thereby limiting cGAS/STING activation.

### Pharmacological activation of mitophagy reduces neuroinflammation and improves neurological function in old mice

Recent studies have highlighted the ability of mitophagy inducers to increase lifespan and healthspan in animal models, as well as their potential as treatments for age-related diseases such as Alzheimer's

disease and sarcopenia[26–28]. Based on our observations indicating upregulation of mitophagy with physiological aging, we hypothesized that pharmacological induction of mitophagy could ameliorate some of the deleterious changes associated with aging (i.e., neuroinflammation and decline of neurological and visual function). We performed an intervention study using the mitophagy inducer urolithin A (UA), a natural metabolite derived from edible plants and fruits such as pomegranate or raspberries, which is already being studied in clinical trials[29,30]. We conducted neurophysiological, behavioral, and retinal immunohistochemical and transcriptomic analyses in young (6 months) and old (22 months) mice injected intraperitoneally with UA (2.3 mg/kg) or vehicle daily for 8 weeks (Fig. 4a). Mass spectrometry (UPLC-ESI-QTOF-MS) of perfused brains, and plasma samples as positive controls, showed that free UA crossed the blood-brain barrier and reached the central nervous system (Fig. 4a). Concentrations of conjugated UA-sulfate were much lower in the brains of UA-treated animals than in plasma samples (Supplementary Fig. 7a), while UA-3-glucuronide was detected only in plasma (Supplementary Fig. 7b), suggesting a possible in situ deconjugating mechanism from

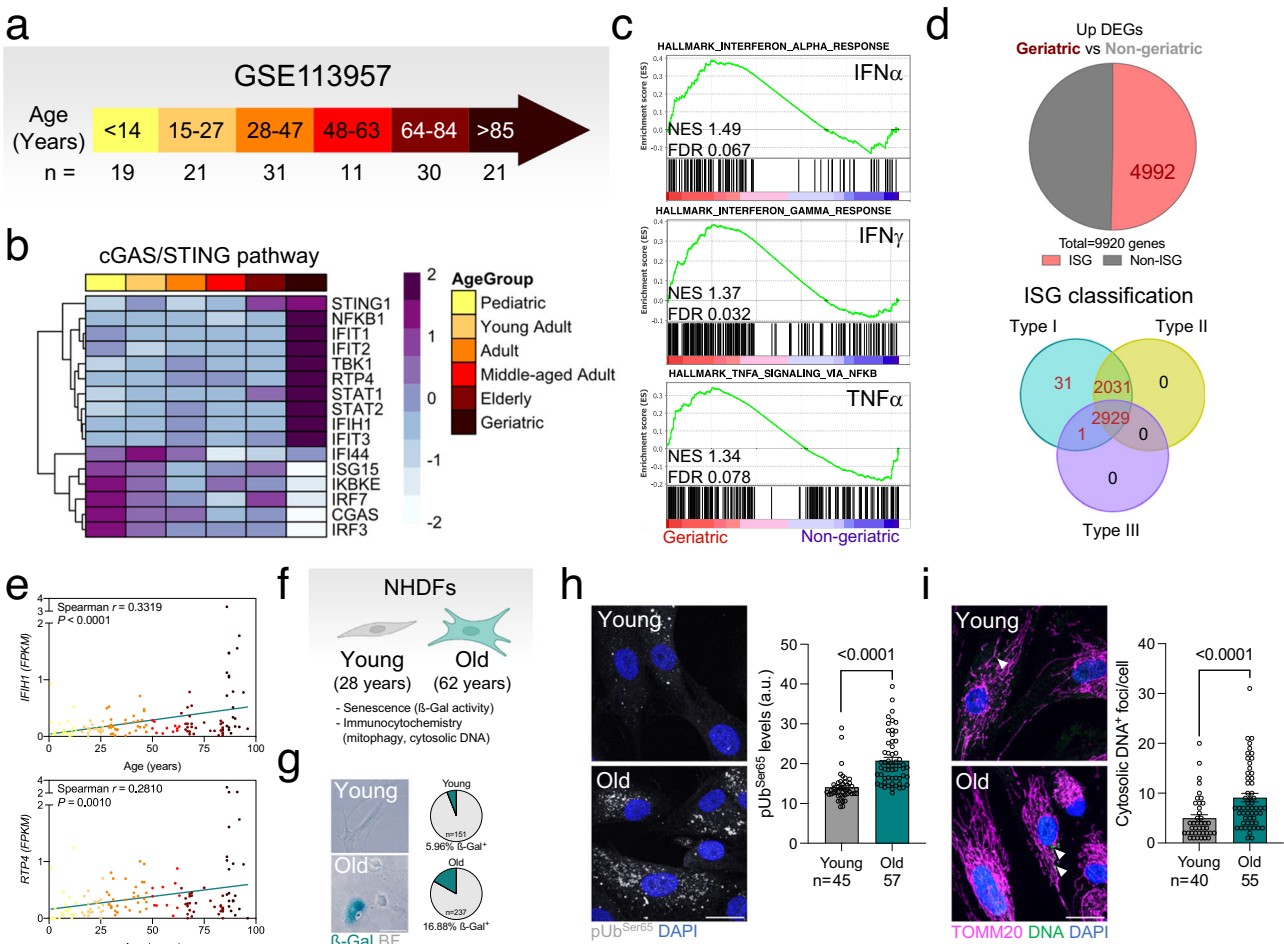

**Fig. 3 | cGAS/STING activation by cytoplasmic DNA is also observed in primary human dermal fibroblasts from elderly donors. a** Age group stratification of primary fibroblast bulk RNAseq data (GSE113957; Fleischer et al.[24]). **b** Heatmap depicting the expression of cGAS/STING pathway modulators across different age groups. **c** Enrichment of IFNα and IFNγ responses and TNFα signaling Hallmarks in geriatric fibroblasts. **d** Interferome analysis (top) showing DEGs that were upregulated in Geriatric fibroblasts, including the proportion of interferon-stimulated genes (ISG, 50.32%); and ISG classification (bottom). **e** Linear regression analysis of mRNA levels of downstream targets of IRF3 (*IFIH1*, *RTP4*) in human fibroblasts during physiological aging (*n* = 133 samples). **f** Experimental design: primary culture of normal human dermal fibroblasts (NHDF) from young (28) and old

(62 years) donors. Created with BioRender.com. **g** Senescence-associated ß-galactosidase activity in young and old fibroblasts (*n* = 151–237 cells). **h** Representative images showing phospho-Ubiquitin$^{Ser65}$ immunostaining (gray) in young and old NHDFs, and corresponding quantification (*n* = 45–57 cells). **i** Representative images showing immunostaining of young and old NHDFs for DNA (green) and TOMM20 (magenta, mitochondria), and corresponding quantification (*n* = 40–55 cells). Scale bars, 25 μm (**g**, **h**, **i**). All data are expressed as the mean ± s.e.m. Dots represent individual samples (**e**) or cells (**h**, **i**). *P* values were calculated using the Spearman rank correlation (**e**) or two-tailed Mann–Whitney *U*-test (**h**, **i**). Created with BioRender.com. Source data are provided as a Source Data file.

UA-sulfate to UA, and an inability of UA-3-glucuronide to cross the low-permeability blood-brain barrier. Since the perfused brain samples were not subjected to enzymatic hydrolysis, this is the first study showing unequivocally the precise Uro metabolic forms that reach the brain. Thus, these results support previous hypotheses suggesting the participation of free UA as a direct effector in brain tissues[31]. UA treatment significantly increased mitophagy in the neuroretina (Fig. 4b) and RPE (Fig. 4c) of young and old mice, albeit to a lesser extent in the latter. Hindlimb clasping score, used as a readout of general neurological function, was significantly lower in old UA-treated mice versus vehicle-treated counterparts (Fig. 4d). Old mice treated with UA treatment also showed improved recognition memory, evaluated using the novel object recognition (NOR) test, compared with vehicle-treated counterparts (Fig. 4e). Rod-mediated twilight-and-night vision deteriorates with age, severely impacting quality of life in the elderly[32]. UA-treated old mice showed improved scotopic (dark) vision, as determined by electroretinogram (ERG) assessment of retinal neurophysiology, suggesting improved rod-mediated visual function (Fig. 4f, g, Supplementary Fig. 8d), as well as increased

expression of genes involved in retinol metabolism (Supplementary Fig. 8e). No differences in mesopic (mixed) and photopic (light) vision were observed in young or old mice treated with UA (Supplementary Fig. 8a–c). Internalization of the rod phototransduction protein Visual Arrestin, an indicator of aberrant integration of light stimuli, was also decreased in old mice treated with UA (Supplementary Fig. 8l). Immunohistochemistry indicated that UA ameliorated CtBP2$^+$mGluR6$^+$ synaptic integrity in old mice (Fig. 4h), further suggesting that UA promotes the preservation of visual function during aging. UA also decreased lipid peroxidation-derived 4-HNE$^+$ aggregates (Fig. 4i), indicating that improved mitochondria quality control may also reduce oxidative stress in the aging retina. Optical coherence tomography (OCT) analysis showed no major morphometric changes (Fig. 4j), and numbers of the main retinal cell types were unchanged (Supplementary Fig. 8h–l). Bulk retina transcriptomic analysis identified age-dependent DEGs associated with UA treatment that were more pronounced in the old group (Supplementary Fig. 9a, b). *k*-means clustering identified 8 specific gene clusters. Genes that were downregulated in old mice (clusters 1, 5, 6, 7) included those involved

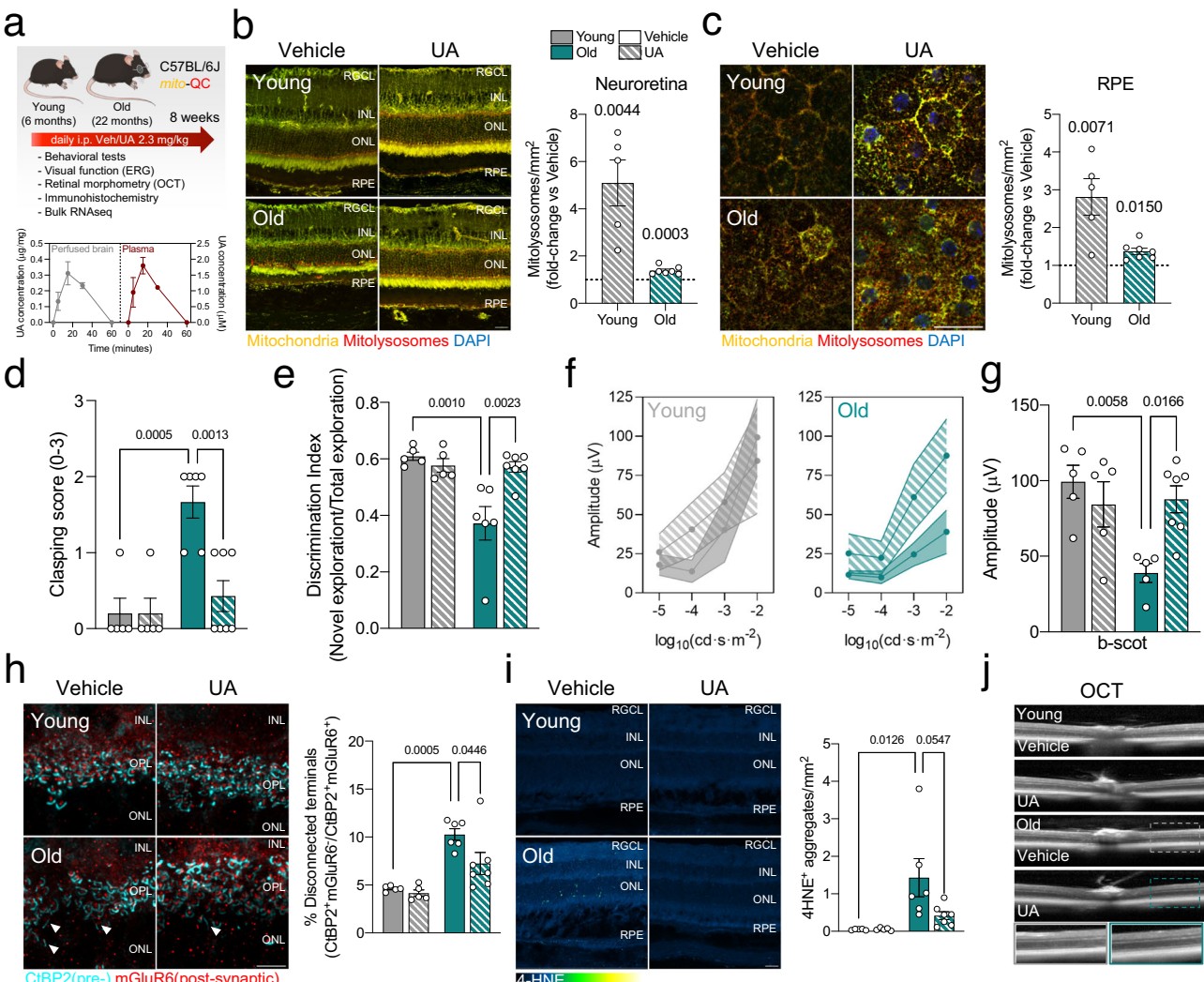

**Fig. 4 | Boosting mitophagy ameliorates age-associated neurological decline.**
**a** Urolithin A (UA, 2.3 mg/kg/day during 8 weeks i.p.) treatment of young and old C57BL/6J *mito*-QC mice: experimental design and readouts (top) and UPLC-ESI-QTOF-MS quantification of the UA levels in perfused brain and plasma (bottom) ($n = 2$–3 mice). Created with BioRender.com. **b** Mitophagy analysis in whole eye cryosections from young and old mice. Representative images (left) and corresponding quantification (right). Dotted line indicates mean in respective vehicle-treated controls ($n = 5$–7 mice). **c** Mitophagy analysis in RPE flatmounts from young and old mice. Dotted line indicates mean in respective vehicle-treated controls ($n = 5$–7 mice). **d** Hindlimb clasping score in young and old mice treated with vehicle or UA ($n = 5$–7 mice). A lower score indicates greater neurological functionality. **e** Discrimination index in the novel object recognition (NOR) test for young and old mice treated with vehicle or UA ($n = 5$–7 mice). **f** Electroretinogram
(ERG) assessment of visual function in scotopic (dark) conditions in young and old mice treated with vehicle or UA ($n = 5$–7 mice). **g** Quantification of b-wave amplitude (µV) at 0.01 cd·s·m⁻² from **f** ($n = 5$–7 mice). **h** Synaptic integrity analysis using immunostaining for pre- (CtBP2, cyan) and post- (mGluR6, red) synaptic terminal markers in young and old mouse retinas: representative images and corresponding quantification ($n = 5$–7 mice). **i** Detection of lipid peroxidation aggregates using anti-4-HNE immunostaining: representative images and quantification ($n = 5$–7 mice). **j** Retinal morphometric assessment by optical coherence tomography (OCT): representative images ($n = 5$–7 mice). Scale bars, 25 µm (**b**, **c**, **i**) and 15 µm (**h**). All data are expressed as the mean ± s.e.m. Dots represent individual mice. *P* values were calculated using a two-tailed Student's *t* test (**b**, **c**) or 2-way ANOVA with Tukey's *post-hoc* test (**d**, **e**, **g**, **h**, **i**). Source data are provided as a Source Data file.

in inflammation and extracellular matrix remodeling, while those that were upregulated (clusters 4, 8) included genes involved in mitochondrial dynamics, metabolism, and visual function (Supplementary Fig. 6c, d).

Crucially, mitophagy stimulation with UA decreased levels of cytosolic DNA (Fig. 5a) and DNA-bound cGAS (Fig. 5a) in old retinas. Furthermore, interferon response (INFα, IFNγ) and TNFα signaling, which were the top 3 positively enriched hallmarks in old mice (Fig. 2b), were also the top 3 negatively enriched hallmarks in UA-treated old mice (Fig. 5b). Transcript levels of cGAS/STING mediators and downstream genes were also reduced in old mice upon induction of mitophagy with UA (Fig. 5c). Retinal aging has also been traditionally associated with an increased glial neuroinflammatory response[33]. UA-treated old mice showed significant reductions in Glial Fibrillary Acidic Protein/GFAP⁺ astrogliosis (Fig. 5d) and a slight tendency towards decreased Ionized calcium-binding adapter molecule 1/Iba1⁺ microglial infiltration (Fig. 5e), suggesting that decreased cGAS/STING signaling may also directly reduce glial activation[34].

UA has also been described to simultaneously stimulate mitochondrial biogenesis in order to restore the healthy mitochondria pool after mitophagy induction[28]. Cytoplasmic mitochondrial mass (*mito*-QC; FIS1-GFP⁺) was indeed increased in the retina of mice treated with UA, a phenomenon that was more pronounced in old mice (Fig. 6a). To further dissect the interplay between cGAS/STING, mitophagy and mitochondrial homeostasis, we set up an in vitro model of mtDNA release using the ARPE-19 cell line[35]. ABT-737 is a Bcl-2 inhibitor that

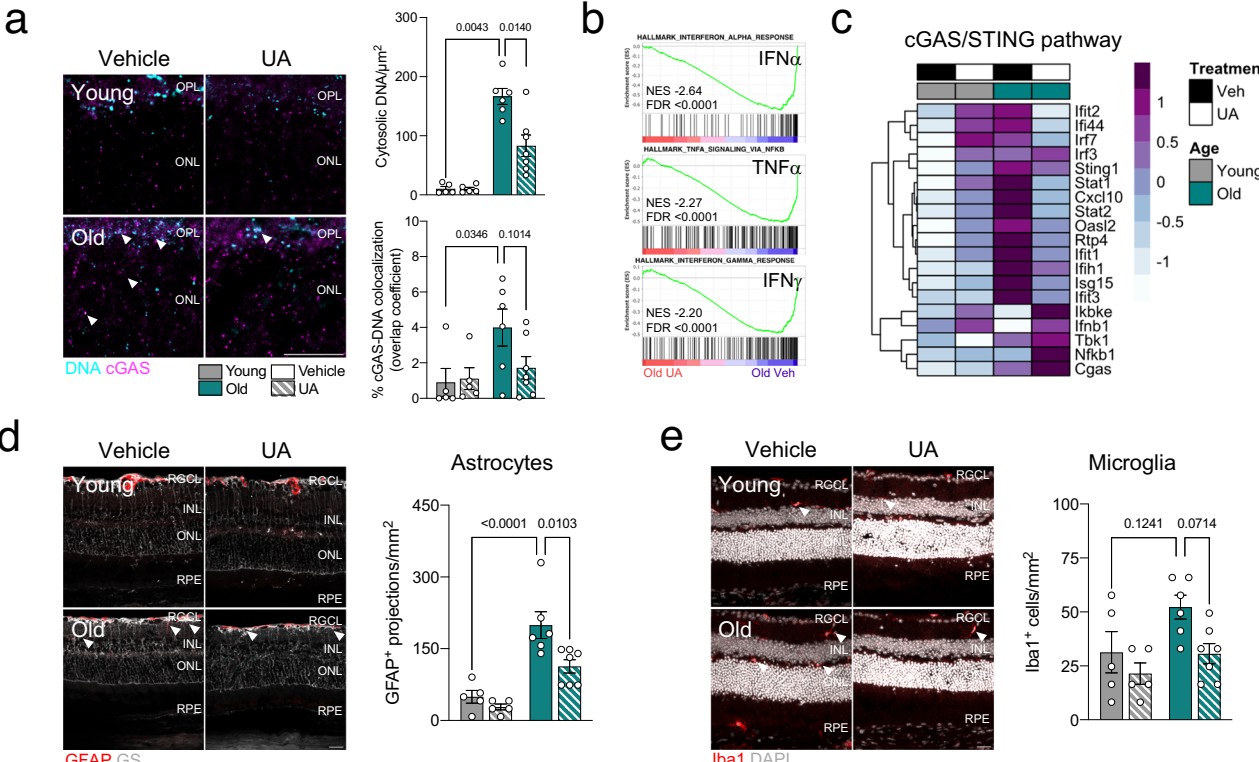

**Fig. 5 | UA treatment attenuates the increased cGAS/STING response observed in old mice. a** Immunostaining for cGAS (magenta) and DNA (cyan): representative images and colocalization analysis (n = 5–7 mice). **b** Top three negatively enriched Hallmarks in old retinas treated with UA (2.3 mg/kg/day) versus vehicle: IFNα and IFNγ responses and TNFα signaling. **c** Heatmap showing expression levels of cGAS/STING pathway modulators in old retinas treated with UA or vehicle. **d** Representative images and quantification of GFAP⁺ astrocytic gliosis (red)

analysis by immunostaining (n = 5–7 mice). Müller glia and astrocytes were counterstained using Glutamine Synthetase (GS). **e** Immunostaining for Iba1⁺ microglial infiltration (red): representative images and corresponding quantification (n = 5–7 mice). Scale bar, 25 µm (**a, d, e**). All data are expressed as the mean ± s.e.m. Dots represent individual mice. *P* values were calculated using the two-tailed Mann–Whitney *U*-test (**a**) or 2-way ANOVA with Tukey's *post-hoc* test (**d, e**). Source data are provided as a Source Data file.

triggers the formation of Bax/Bak pores in the outer mitochondrial membrane, leading to the release of its inner components including mtDNA[18]. Cells were also treated with the pan-caspase inhibitor Q-VD-OPh (QVD) to avoid apoptosis induction due to cytochrome c release (Fig. 6b). Treatment with ABT-737 increased the number of cytosolic DNA foci, a phenomenon that was fully reverted when the cells were co-treated with UA but not with the selective cGAS inhibitor G140 (Fig. 6c)[36]. ABT-737 also stimulated the cGAS/STING signaling cascade and this effect was abolished by both UA and G140 (Fig. 6d). Taken together, these data validate that UA reduces cGAS/STING-mediated inflammation by promoting mitochondria quality control and eliminating its triggering event (mtDNA release) rather than modulating downstream signaling. Markedly, ABT-737-induced mitophagy was reduced when cells were co-treated with G140 indicating that mitophagy is indeed a cytoprotective response to cGAS/STING activation (Fig. 6e). Reproducing our findings on old C57BL/6J mice (Fig. 1c, d) and NHDFs from old donors (Fig. 3h), ABT-737 significantly increased phospho-Ubiquitin^Ser65 levels (Supplementary Fig. 10a) pointing to activation of PINK1/Parkin-dependent mitophagy. Similarly, no changes were observed in receptor-mediated mitophagy effectors (Supplementary Fig. 10b), cardiolipin levels or its translocation to the OMM (Supplementary Fig. 10c). Mimicking our findings in mice retina, UA also induced mitochondrial biogenesis in the ABT-737 model and slightly reduced mitochondrial ROS production (Fig. 6f). Most notably, mitochondrial respirometry analysis revealed that UA restored basal respiration and ATP production in ABT-737-treated cells and significantly improved spare respiratory capacity (Fig. 6g).

Finally, to fully ascertain whether mitophagy is the sole responsible of UA-derived beneficial effects or whether mitochondrial

biogenesis is also involved, we evaluated the same readouts in ARPE-19 cells where PINK1/Parkin-dependent mitophagy (*PINK1;PARK2*-siRNA; Fig. 7f) and/or mitochondrial biogenesis (100 µM Chloramphenicol; Fig. 7c) were simultaneously downregulated. PINK1/Parkin knockdown efficiently abolished mitophagy induction by both ABT-737 and UA (Fig. 7a, b). Mitochondrial biogenesis inhibition had no effect on mitophagy levels (Fig. 7a, b) nor cytosolic DNA decrease induced by UA (Fig. 7d, e). However, PINK1/Parkin knockdown fully abrogated the UA-mediated decrease on cytosolic DNA foci in ABT-737-treated cells (Fig. 7e, f) indicating that this pathway is indeed responsible for curtailing mtDNA release.

Together, these data constitute proof of concept supporting the therapeutic potential of pharmacological induction of mitophagy to attenuate age-associated inflammation and neurological deterioration.

## Discussion

Age is the primary risk factor for many illnesses, including Alzheimer's disease, age-related macular degeneration and cancer, and population aging is leading to substantial increases in the socioeconomic burden of many diseases[37]. While the etiology of age-associated diseases is gradually being elucidated, the primary mechanisms that drive aging remain unclear[4,5]. Our findings demonstrate for the first time age-associated physiological upregulation of mtDNA-cGAS/STING-mediated inflammation, which is conserved across different organs and species and can be curtailed by the induction of mitophagy (Fig. 8). Here, we describe crosstalk between mitochondrial quality control and inflammation that may help explain the lifespan-extending effects of several strategies based on the activation of autophagy[38,39] and mitophagy[28,40].

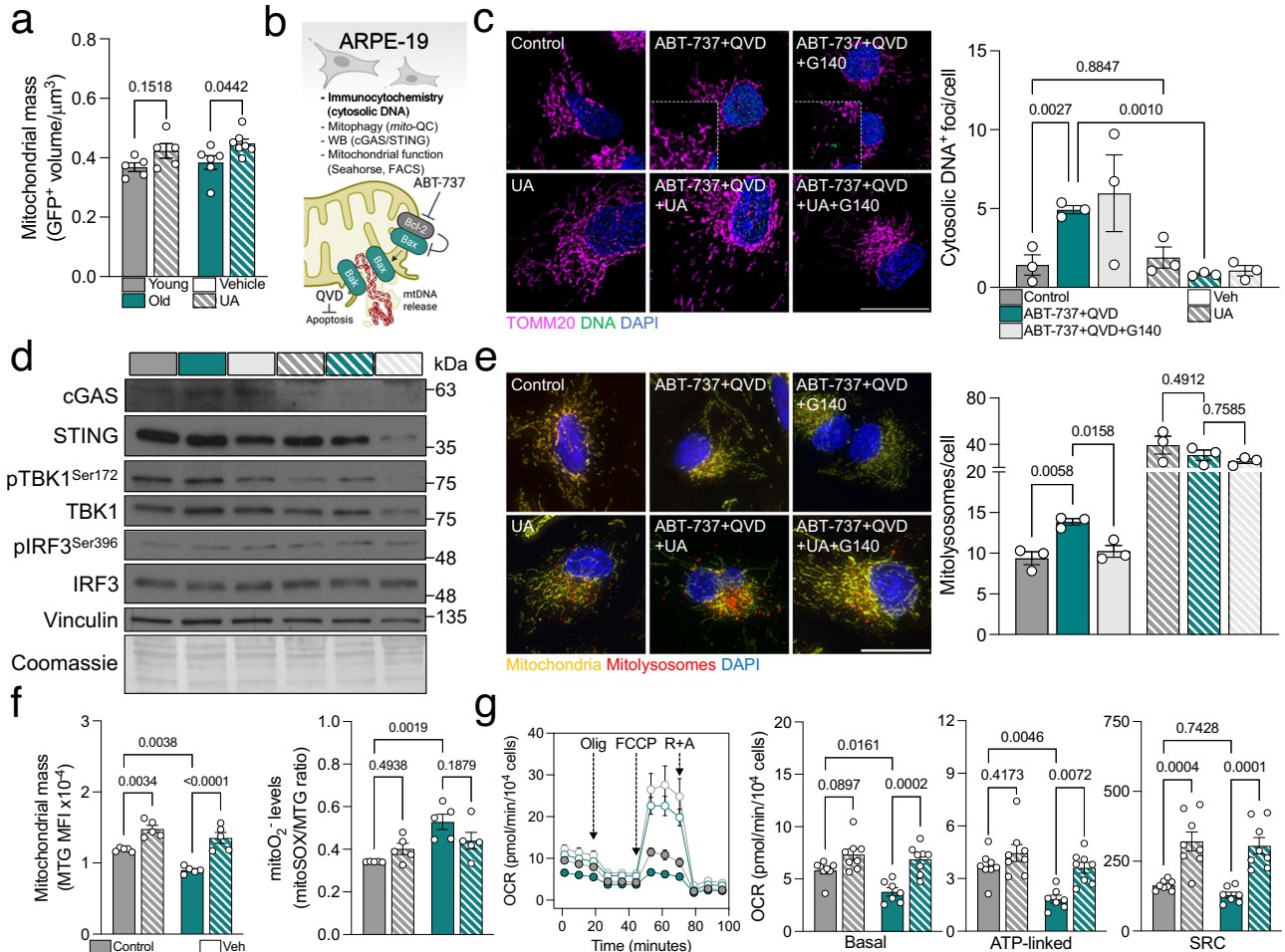

**Fig. 6 | UA induces mitochondrial biogenesis and improves mitochondrial function. a** Analysis of mitochondrial mass (GFP+ volume) in the retina of young or old *mito*-QC mice treated with UA or vehicle as shown in Fig. 4a, b (*n* = 5–7 mice). **b** Experimental design and readouts for the in vitro model of mtDNA release using the ARPE-19 cell line and the Bcl-2 inhibitor ABT-737 (10 µM). Pan-caspase inhibitor Q-VD-OPh (QVD, 10 µM) was added to inhibit apoptosis. Created with BioRender.com. **c** Representative images showing immunostaining of ARPE-19 cells for DNA (green) and TOMM20 (magenta, mitochondria), and corresponding quantification. UA (100 µM) was used as a mitophagy inducer and human cGAS inhibitor G140 (10 µM) was added as a control. (*n* = 3 independent experiments) **d** Western blot analysis of cGAS/STING mediators (TBK1, IRF3, cGAS, STING) in ARPE-19 cells (*n* = representative of 3 independent experiments). Protein levels are normalized to the loading control (Vinculin). **e** Representative images of mitophagy analysis in

ARPE-19 cells using *mito*-QC reporter and corresponding analysis (*n* = 3 independent experiments). **f** Analysis of mitochondrial mass (MitoTracker Green; MTG) and mitochondrial superoxide production (MitoSOX Red/MTG ratio) by flow cytometry in ARPE-19 cells (*n* = 5 independent experiments). **g** Mitochondrial respirometry analysis in ARPE-19 cells using Seahorse XFe24 after sequential injection of Oligomycin, FCCP and Rotenone+Antimycin (*n* = 7–8 biological replicates from two independent assays). Oxygen consumption rate was normalized to cell number and graphs show basal respiration, ATP-linked respiration and spare respiratory capacity (SRC). Scale bars, 25 µm. All data are expressed as the mean ± s.e.m. Dots represent individual mice (**a**), independent experiments (**c, e, f**) or biological replicates from two independent assays (**g**). *P* values were calculated using 2-way ANOVA (**a**) or 1-way ANOVA with Šídák's (**c, e**) or Tukey's (**f, g**) *post-hoc* test (**c, e**). Source data are provided as a Source Data file.

Previous research has shown that PINK1-dependent mitophagy is dispensable to sustain basal mitophagy in young mice[41]. Supporting these data, we also observed low levels of PINK1-mediated Ubiquitin[Ser65] phosphorylation in all organs analyzed, in stark contrast with the very high levels observed in old mice and primary fibroblasts from aged donors. Similar findings have been described in post-mortem aged human brain samples[11]. While no studies focused on mitophagy or neuroinflammation have been conducted using aged PINK1 or Parkin-deficient mice, young *PINK1*−/− animals already show slight mitochondrial function defects that are further heightened in old mice suggesting a key role of this pathway in maintaining mitochondrial health upon aging[42,43]. Young mice lacking PINK1 or Parkin also show an exacerbated inflammatory response to external stress, a phenotype that is fully abolished when STING is also absent[44]. Mitophagy has also been found to be increased in neurons and flight muscle of the model organism *Drosophila melanogaster* and this phenotype

was fully abrogated in PINK1 and parkin-deficient flies[9], further reinforcing the cross-species nature of our findings in mammals. Most importantly, the motor phenotype of Parkin-deficient flies was again alleviated in animals that simultaneously lack STING[45]. Interestingly, the R293Q STING variant, which yields a less functional version of the protein, has been associated with decreased risk of developing age-associated diseases in large cohorts of elderly[34,46].

While we observed a clear correlation between increased mitophagy, mtDNA release and cGAS/STING activation upon aging, why basal mitophagy cannot fully restrict age-related neuroinflammation remains puzzling. In our in vivo and in vitro models, treatment with UA simultaneously induced mitophagy −even though to a lesser extent in old mice− and mitochondrial biogenesis. UA has been previously reported to increase mitochondrial mass by targeting the Peroxisome proliferator-activated receptor Gamma Coactivator 1-alpha (PGC-1α)[30,47] and Nuclear factor erythroid 2-Related Factor 2 (NRF2)

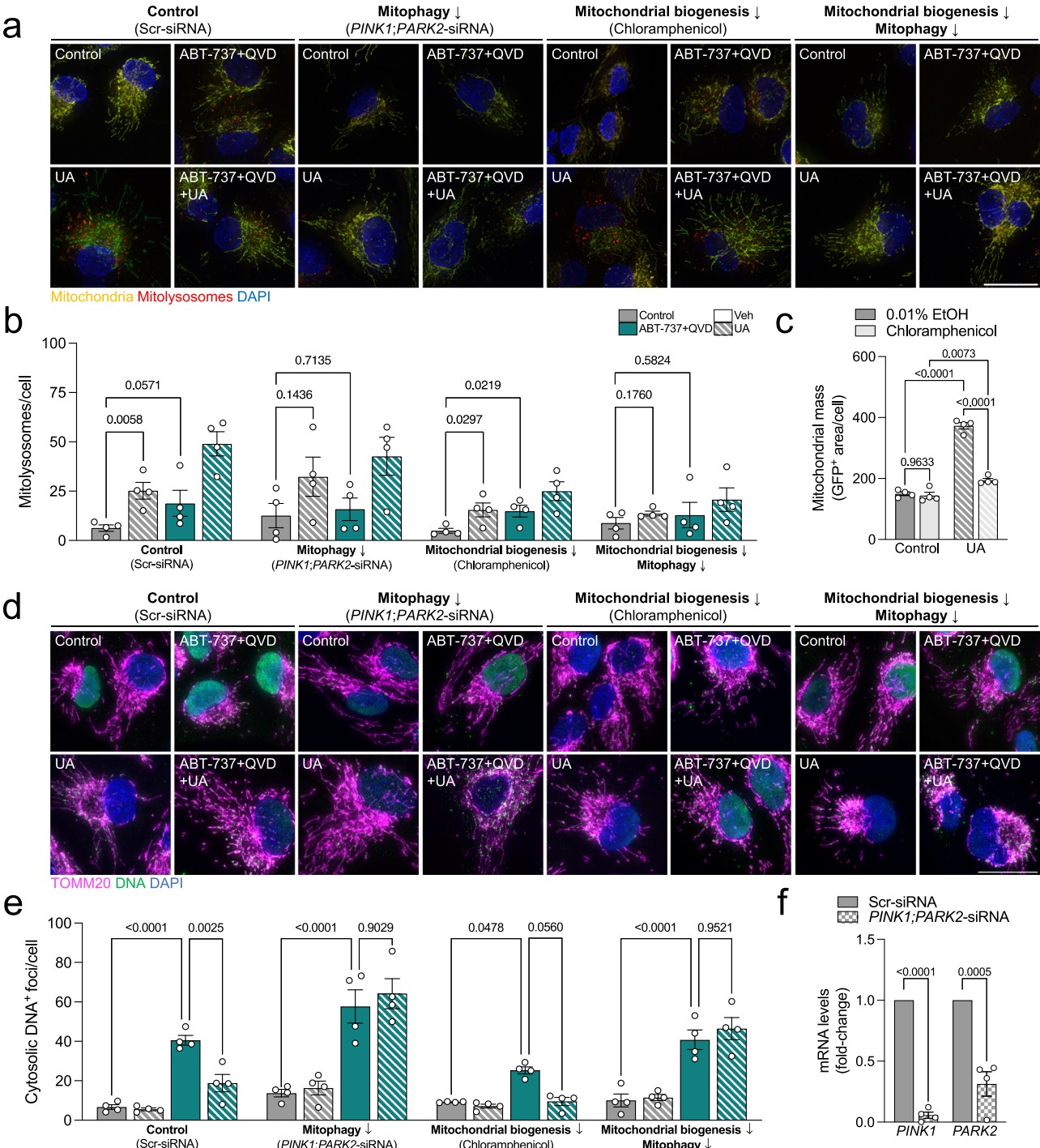

**Fig. 7 | PINK1/Parkin-dependent mitophagy stimulation by UA mediates cytosolic ABT-737-induced DNA decrease. a** Representative images of ARPE-19 cells expressing the *mito*-QC reporter which subjected to PINK1/Parkin-depedent mitophagy (*PINK1; PARK2* knockdown) and/or mitochondrial biogenesis (100 μM Chloramphenicol) inhibition, and simultaneously treated with ABT-737 and/or UA. **b** Quantification of the number of mitolysosomes per cell as shown in Fig. 7a (*n* = 4 independent experiments). **c** Quantification of mitochondrial mass in the presence or absence of Chloramphenicol to validate UA-induced mitochondrial biogenesis inhibition, as reported in Fig. 4a, f (*n* = 4 independent experiments). **d** Representative images showing immunostaining of ARPE-19 cells for DNA (green) and TOMM20 (magenta, mitochondria). **e** Quantification of the number of cytosolic DNA foci per cell as shown in Fig. 7d (*n* = 4 independent experiments). **f** Quantification of *PINK1* and *PARK2* mRNA levels to validate siRNA-mediated knockdown efficiency (*n* = 4 independent experiments). Scale bars, 25 μm. All data are expressed as the mean ± s.e.m. Dots represent independent experiments. *P* values were calculated using two-tailed Student's *t* test (**b**, **c**) or two-tailed Mann–Whitney's *U*-test (**b**, Control:Control vs Control:ABT), 1-way ANOVA with Šídák's *post-hoc* test (**e**) or 2-way ANOVA with Šídák's *post-hoc* test (**f**). Source data are provided as a Source Data file.

pathways[28,48] in other experimental settings. Therefore, tuning down cGAS/STING-mediated inflammation long term could require both aspects of mitochondrial quality control: removing damaged components and supplying new ones. Supporting this hypothesis, it has been shown that boosting mitochondrial biogenesis through PGC-1α overexpression decreased PINK1/Parkin-mediated mitophagy in aged skeletal muscle and improved mitochondrial function[10]. In a recent randomized trial in healthy middle-aged adults, oral supplementation

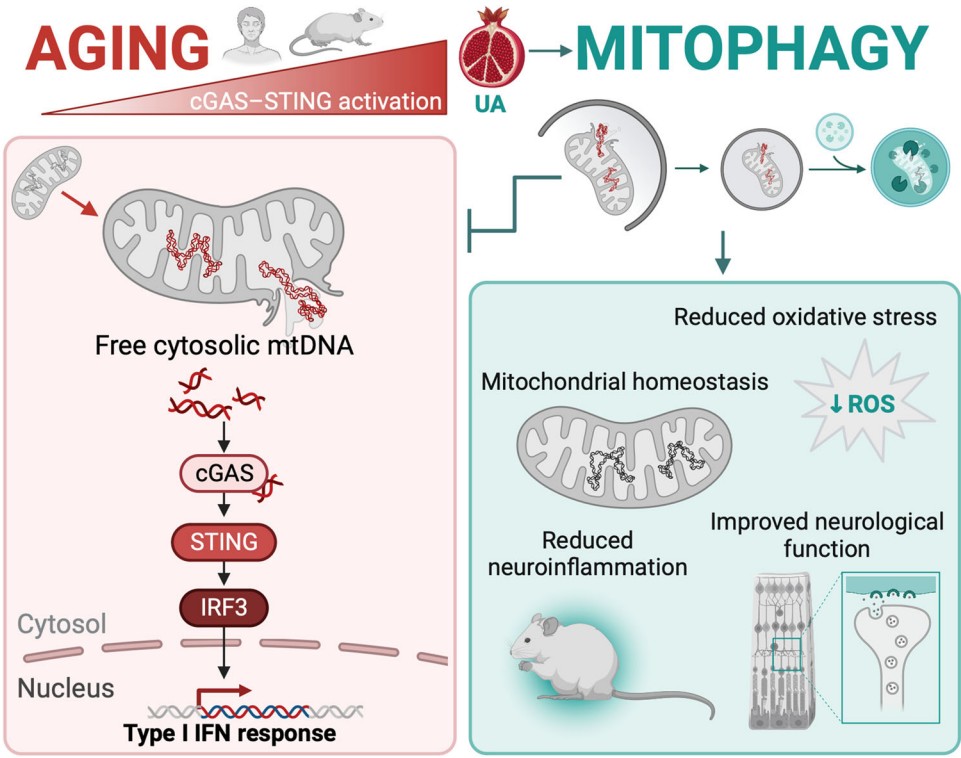

**Fig. 8 | cGAS/STING-driven neuroinflammation associated with aging is modulated by mitophagy.** Age-associated mitochondrial malfunction leads to mtDNA release triggering cGAS/STING activation and type I IFN response in mice and humans. Restoring mitochondrial homeostasis using UA constrained mtDNA leakage, reduced inflammation and improved healthspan. Created with BioRender.com.

with UA reduced serum levels of inflammatory marker C-reactive protein (CRP) and promoted mitochondrial biogenesis and homeostasis[30] highlighting its potential as a dietary supplement to promote healthy aging.

A recent report has shown that treatment with STING inhibitors can reduce age-associated neuroinflammation, neurotoxicity and improve cognition[49], thus supporting our findings on cGAS/STING activation driving physiological decline. However, the long-term effects of cytosolic DNA sensing inhibition might render the immune system more susceptible to pathogen infection and limit its translational prospect in the context of aging. The possibility of indirectly reducing age-associated inflammation and disease through mitophagy induction constitutes a promising therapeutic approach that warrants further exploration.

## Methods
### Animal models
All animal experiments were performed according to RU guidelines and the ARVO Statement for the Use of Animals in Ophthalmic and Vision Research, and were authorized by the institutional bioethics committee at Centro de Investigaciones Biologicas Margarita Salas (CSIC) and Comunidad de Madrid (PROEX 154.3/21). C57BL/6J *mito*-QC mitophagy reporter mice were generated and kindly provided by Ian Ganley (University of Dundee, Scotland)[14]. Briefly, targeted transgenesis was used to induce a constitutive knock-in of the fusion protein mCherry-GFP-FIS1[101-152] in the *Rosa26* locus. This construct includes a pH-insensitive fluorescent protein (mCherry), a pH-sensitive fluorescent protein (GFP), and the mitochondria-targeting sequence of FIS1. Mice were bred and housed at the CIB animal facility, on a 12/12 h light/dark cycle with *ad libitum* access to food (standard chow) and water. Mice were classified as young (6–8 months) or old (22–26 months). Both male and female mice were included in all experiments. For select experiments, mice were intraperitoneally injected with either 40 mg/kg Leupeptin (L2884, Merck) or saline 16 h before sample collection. All animals were sacrificed at 9:00 a.m. to avoid circadian variations in autophagy, metabolism, and the visual cycle. The colony was routinely genotyped by end-point PCR to ensure homozygosity using the following primers: set 1, 5′-CAAAGACCCCAACGAGAAGC-3′ and 5′-CCCAAGGCACACAAAAAACC-3′; set 2, 5′-CTCTTCCCTC GTGATCTGCAACTCC-3′ and 5′-CATGTCTTTAATCTACCTCGATGG-3′.

### Primary human cell culture
Normal human dermal fibroblasts (NHDFs), isolated from adult abdominal dermal explants from 28-year-old (Young, DFM050410A) and 68-year-old (Old, DFM050510B) donors, were purchased from ZenBio. Both cell lines had been previously tested negative for HIV-1, HIV-2, HTLV-1, HTLV2, hepatitis B, hepatitis C, and mycoplasma. Fibroblast morphology and viability were also validated. Cells were maintained in high-glucose DMEM (41966-029, Gibco) supplemented with 15% FBS (F7524, Merck), 2 mM L-glutamine (25030081, Gibco), and 1 U/mL Pen/Strep antibiotics (15140122, Gibco), and stored in a humidified incubator at 37 °C, 5% $CO_2$. For immunofluorescence analysis, $5 \times 10^4$ cells were seeded on glass coverslips in a 24-well plate. For western blot analysis, $3 \times 10^5$ cells were seeded in 6-well plates.

### ARPE-19 cell line culture
ARPE-19 cells (ATCC, CRL-2302) were maintained in 1:1 high-glucose DMEM and F12 (21765037, Gibco) supplemented with 15% FBS (F7524, Merck), 2 mM L-glutamine (25030081, Gibco), and 1 U/mL Pen/Strep antibiotics (15140122, Gibco), and stored in a humidified incubator at 37 °C, 5% $CO_2$. Cells stably expressing the *mito*-QC reporter were generated in the laboratory of Dr. Ian Ganley[50] and selected using 800 µg/mL Hygromycin B (10453982, Gibco). For Seahorse, FACS and immunofluorescence analysis, $5 \times 10^4$ cells were seeded in a 24-well plate. For western blot analysis, $3 \times 10^5$ cells were seeded in 6-well plates. Whenever indicated, cells were treated with 100 µM UA (6762,

Tocris), 10 µM ABT-737 (S1002, Selleckchem), 10 µM Q-VD-OPh (S7311, Selleckchem), 10 µM G140 (inh-g140, Invivogen) or 100 µM Chloramphenicol (C0378, Merck) for 24 h. Gene knockdown was achieved by transfecting the cells for 24 h with Silencer Select Pre-Designed siRNAs (PINK1, s35166; PARK2, s530998, Thermo Fisher) and Lipofectamine RNAiMAX (13778075, Thermo Fisher) following manufacturer's instructions. Knockdown efficiency was validated by RT-qPCR using TaqMan probes (Thermo Fisher) targeting *PINK1* (Hs00260868_m1) or *PARK2* (Hs01038318_m1), and ribosomal *18S* (Hs99999901_s1) was used as reference gene.

### Tissue processing and mitophagy assessment using *mito*-QC

Mice were deeply anesthetized using ketamine/xylazine (80 mg/kg and 10 mg/kg, respectively) and transcardially perfused with 50 mL PBS pH 7.0. Distinct tissue types were dissected and fixed by immersion for 4 h in 3.7% PFA (50-980-487, EMS) containing 175 mM HEPES (15630056, Gibco) pH 7.0. Samples were cryoprotected in 30% sucrose (1.07651.1000, Millipore) and embedded in optimal cutting temperature (OCT) compound (4583, Sakura). Cryosections (12 µm-thick) were cut using a Leica CM1950 cryostat and stored at −20 °C. Brains were flash frozen and 40-µm free-floating sections were stored at 4 °C. For mitophagy assessment, samples were air-dried for 5 min and rehydrated with PBS pH 7.0 before incubation with 1 µg/mL DAPI (D9542, Merck) in PBS pH 7.0 for 15 min. Slides were washed 3 times with PBS pH 7.0 and mounted using VECTASHIELD (H-1000-10, Vector Laboratories). For mitolysosome quantification, images were captured with a 0.5-µm z-step using a Leica TCS SP8 confocal microscope equipped with a 63× immersion objective.

### Immunofluorescence and immunocytochemistry

Tissue cryosections were permeabilized by immersion for 15 min in 0.3% Triton X-100 in PBS pH 7.0 (T9284, Merck) and blocked for 1 h with block/perm buffer (10% NGS (G9023, Merck), 0.1% Triton X-100 in PBS pH 7.0). Sections were incubated overnight at 4 °C with primary antibodies at the indicated dilution (Supplementary Table 1) in block/perm buffer. The following day, sections were incubated for 1 h at room temperature with fluorophore-conjugated secondary antibodies diluted 1:200 in block/perm buffer. Cardiolipin was labeled using 5 µM Nonyl Acridine Orange (NAO; A1372, Thermo Fisher) for 1 h at room temperature. Nuclei were counterstained with 1 µg/mL DAPI for 15 min. Sections were washed 3 times with PBS pH 7.0 after each step. Sections were mounted using VECTASHIELD or Fluoromount-G (100-01, Bionova) and imaged with a 1 µm z-step using a Leica TCS SP8 confocal microscope equipped with a 63× immersion objective. Immunocytochemistry was performed using BGT (3% BSA (MB04602, NZYtech), 0.25% Triton X-100 in PBS pH 7.0) and antibodies were incubated for 1 h at room temperature. Samples were mounted using ProLong Diamond (P36961, Thermo Fisher) and imaged using a Leica Thunder epifluorescence microscope equipped with a 63× immersion objective. β-gal assay was performed using the Senescence β-galactosidase Staining Kit (9860, Cell Signaling) following the manufacturer's instructions and images were acquired with a bright field microscope.

### Subcellular fractionation and qPCR

Cytosolic, nuclear, and mitochondrial fractions were isolated from mouse retinas as previously described[51]. DNA was isolated using phenol:chloroform. Briefly, fractions were resuspended in TRIzol reagent (15596018, Thermo Fisher) and incubated for 5 min at room temperature. Chloroform (1:5 Chlor:TRIzol) was added and fractions were incubated for a further 3 min. Samples were centrifuged for 15 min at 12,000 × g at 4 °C. The RNA-containing fraction was discarded and pure ethanol (1:3.3 EtOH:TRIzol) was added to the solution, which was mixed by inversion and incubated for 3 min. Samples were centrifuged for 5 min at 2000 × g at 4 °C to pellet the DNA and the supernatant was discarded. Next, the DNA was resuspended in SC buffer (0.1 M sodium citrate in 10% EtOH, pH 8.5) and incubated for 30 min, mixing every 5 min by inversion. Samples were centrifuged for 5 min at 2000 × g at 4 °C and the supernatant was discarded twice to wash the DNA. The pellet was resuspended in 75% ethanol and incubated for 15 min, mixing every 5 min by inversion, and centrifuged for 5 min at 2000 × g at 4 °C. The DNA was air-dried and resuspended in NE buffer (MB135, NZYtech). Quality and quantity of each sample was assessed using a NanoDrop 1000 (Thermo Fisher). Final DNA was diluted 1:10 (final 0.25 µL per well in 10 µL reaction volume) and quantitative RT-qPCR was performed using a LightCycler 480 thermocycler (Roche). The following TaqMan gene expression probes were used: *mt-Nd2* (Mm04225288_s1), *mt-Co1* (Mm04225243_g1), *mt-Cytb* (Mm04225271_g1), and *18S* (Hs99999901_s). Results were normalized to nuclear *18S* expression levels and the cytosolic/mitochondrial ratio of mtDNA-encoded genes was determined.

### Protein isolation and western blot

Adherent cells were scraped and tissues were homogenized in cold RIPA lysis buffer (R0278, Merck) supplemented with protease and phosphatase inhibitors using an IKA T8 Ultra Turrax tissue homogenizer (Cole-Parmer). Protein concentration was determined using the Pierce BCA Protein Assay (23225, Thermo Scientific) following the manufacturer's instructions. Total protein extract (12–45 µg) was supplemented with 5X loading buffer (4% glycerol, 0.5 M Tris-HCl pH 6.8, 8% SDS, 0.04% bromophenol blue, 5% β-mercaptoethanol) and resolved on Any kD Criterion TGX Precast Stain-free gels (5678124, Bio-Rad). Proteins were transferred to 0.2 µm PVDF membranes using a TransBlot Turbo Transfer System (Bio-Rad) and total protein was quantified using Ponceau S staining (78376, Merck). Membranes were blocked with 5% non-fat milk in PBS-T (0.5% Tween-20 [1706531, Bio-Rad] in PBS) for 1 h. Membranes were then incubated overnight at 4 °C in primary antibodies diluted 1:1000 (Supplementary Table 1) in 5% BSA in PBS, and subsequently for 1 h at room temperature in secondary antibodies diluted 1:2000–1:4000 in PBS-T. Membranes were developed using Pierce ECL Western Blotting substrate (32106, Thermo Fisher) or Amersham ECL Prime (10308449, Cytiva) and x-ray film (AGFA) using a CURIX 60 Processor (AGFA).

### Urolithin metabolite determination

Urolithin A 3-*O*-glucuronide (Uro-A 3-glur, UA-3-glucuronide), urolithin B 3-*O*-glucuronide (Uro-B glur), urolithin A sulfate (Uro-A sulfate, UA-sulfate), urolithin B 3-*O*-sulfate (Uro-B sulfate), urolithin A (Uro-A, UA), and urolithin B (Uro-B) were chemically synthesized and purified by Villapharma Research S.L. (Parque Tecnológico de Fuente Alamo, Murcia, Spain). Stock solutions (10 mM) of individual urolithins were prepared in dimethyl sulfoxide (DMSO), and standard mixtures were prepared in methanol at 200 µM. 6,7-dihydroxycoumarin (246573, Sigma-Aldrich) was used as an internal standard, and the stock solution was prepared in methanol. All standard solutions were stored at −20 °C.

 *mito*-QC mice were intraperitoneally injected with PBS (*n* = 3) or 5 mg/kg Uro-A (*n* = 9) (a human equivalent dose of approximately 30 mg Uro-A in a 70 kg person)[52]. PBS-treated mice were used as controls. First, blood was collected from the submandibular plexus. Mice were perfused with PBS to remove residual blood from the brain immediately after blood collection at each post-administration time-point (5, 15, 30, and 60 min). Frozen perfused brains were ground in liquid $N_2$ to obtain the finely powdered samples. Next, 350 mg of each tissue sample was mixed with 1000 µL of 2% formic acid in acetonitrile (100030, Sigma-Aldrich) for protein precipitation. Next, samples were homogenized for 5 min in a Bullet Blender Tissue Homogenizer (Next Advance Inc.), sonicated in a bath for 10 min, and centrifuged at 16,000 × g for 5 min at 4 °C. A second extraction was performed using 1 mL of the same acidified acetonitrile. Next, both supernatants were pooled and reduced to dryness

using a Speedvac concentrator (Thermo Savant SPD121P, Thermo Fisher Scientific). Finally, the pellet was re-dissolved in 100 μL methanol:water:formic acid (49.9:50:0.1, $v/v/v$), centrifuged at $16,000 \times g$ for 5 min, and filtered through a 0.2 μm, 4 mm PTFE filter before analysis by UPLC-ESI-QTOF-MS. Plasma was obtained by centrifugation at $1000 \times g$ for 10 min at 4 °C and immediately frozen at −80 °C. Next, plasma samples (approximately 100 μL) were extracted with acetonitrile:formic acid (98:2, v/v) (ratio plasma/acetonitrile:formic acid 1/3). After centrifugation for 10 min at $14,000 \times g$ and 4 °C, the supernatant was evaporated in the speed vacuum concentrator. Finally, the samples were re-dissolved in 100 μL of methanol, filtered through a 0.2 μm, 4 mm PTFE filter, and analyzed by UPLC-ESI-QTOF-MS.

Uro-A and its derived conjugated metabolites were determined by UPLC-ESI-QTOF-MS. The instrument consisted of an Agilent 1290 Infinity UPLC system coupled to a 6550 Accurate-Mass Quadrupole time-of-flight (QTOF) mass spectrometer (Agilent Technologies). Separation was carried out in a Poroshell 120 EC-C18 column ($3 \times 100$ mm, 2.7 μm) (Agilent) at 25 °C. Water:formic acid (99.9:0.1, v/v) and acetonitrile:formic acid (99.9:0.1, v/v) were used as mobile phases A and B, respectively, with a flow rate of 0.4 mL/min, and the same gradient and chromatographic conditions as previously described[53]. Before injection, 0.2 ppm of an internal standard (6,7-dihydroxycoumarin) was added to the samples. Data were processed using Mass Hunter Qualitative Analysis software (version B.08.00, Agilent). Compounds were determined by directly comparing MS spectra with available standards and confirmed based on their spectral properties, molecular mass, and fragmentation pattern as previously reported[53]. Uro-A and derived metabolites were quantified by interpolating the calibration curve obtained with their corresponding standards. Finally, compounds were quantified in MS by peak area integration of corresponding extracted ion chromatograms[53].

### Intervention study with UA

Young (6 months) and old (22 months) *mito*-QC mice were injected intraperitoneally with 2.3 mg/kg/day UA or vehicle (DMSO) for 8 weeks. All animals received a health check by the head veterinarian at the CIB Animal Facility before the beginning of the study to rule out any confounding pathological conditions. At the end of the treatment period the animals underwent behavioral and electrophysiological evaluation and were sacrificed by cervical dislocation the day after the last injection.

### Behavioral tests

**Limb clasping.** Mice were suspended by the tail for 10 s and clasping scored on a scale of 1–3, as previously reported[54].

**Novel object recognition (NOP).** Short-term (3 h) memory was assessed using the NOP test. Mice were trained in an open field arena with two identical objects for 5 min, followed by a 3 h retention period. One of the objects was replaced by a novel object and mice were placed in the same arena for another 5 min. Interaction time with each object was quantified and the discrimination index was calculated as the ratio between novel object exploration and total object exploration[54]. All tests were recorded and subsequently evaluated by an investigator blind to age group and treatment.

### Electroretinogram (ERG) recording and optical coherence tomography (OCT)

Mice were dark-adapted overnight and all manipulations were performed under low-intensity red light. Anesthesia was achieved by intraperitoneal injection (50 mg/kg ketamine, 0.3 mg/kg medetomidine) and pupils were dilated using 1% tropicamide. Mice were placed on a heating pad at 37 °C for the duration of the procedure. A drop of 2% methocel (Omnivision GmbH) was added to each eye and corneal electrodes were placed parallel to the corneal surface. A reference electrode was placed in the mouth and a ground electrode on the tail. Flash-induced ERG responses were recorded in response to increasing light stimuli produced by a calibrated Ganzfeld stimulator, and 5–20 readings were averaged for each intensity. Scotopic (0.00001, 0.0001, 0.001, 0.01 cd·s·m$^{-2}$) and mixed (0.1, 0.32, 1, 3.2, 10, 32 cd·s·m$^{-2}$) responses were recorded in dark conditions. After light adaptation for 5 min, photopic responses (0.1, 0.32, 1, 3.2, 10, 32 cd·s·m$^{-2}$) and flicker (32 cd·s·m$^{-2}$) were recorded. ERG signals were amplified and band-filtered between 0.3 and 1000 Hz with an amplifier (CP511 AC amplifier; Grass Instruments). Electrical signals were digitized at 20 kHz with a power laboratory data acquisition board (AD Instruments). Wave amplitude was measured using LabChart 7 (AD Instruments). OCT analysis was performed immediately after ERG recording using a SPECTRALIS ophthalmic imaging platform (Heidelberg Engineering). A contact lens was placed on top of the cornea and images were captured in high-resolution mode. Both eyes were analyzed and no differences were detected between left and right eyes.

### Transmission electron microscopy

Samples were prepared previously described[6]. Briefly, whole eyes were fixed for 4 h at 4 °C in Karnovsky fixative solution (4% paraformaldehyde (w:v), 2.5% glutaraldehyde (v:v) in 0.1 M sodium phosphate buffer, pH 7.14), and washed and stained with 5% osmium tetroxide (w:v) prior to embedding in epoxy resin (EPON, Hexion). Ultra-thin sections (50 nm) were cut on a Vitracut E ultramicrotome (Reichert-Jung), stained with uranyl acetate and lead citrate, and imaged with a Zeiss EM 902 transmission electron microscope at 90 kV.

### High-throughput bulk RNAseq and bioinformatic analysis

Total RNA extractions were quantified with a Nanodrop One device (Thermo Fisher), and RNA integrity was assessed using the Bioanalyzer 2100 RNA Nano assay (Agilent). Libraries for RNA-seq were prepared at the Functional Genomics Core Facility, IRB Barcelona. Briefly, mRNA was isolated from 620 ng total RNA using the NEBNext Poly(A) mRNA Magnetic Isolation Module (E7490L, New England Biolabs). Isolated mRNA was used to generate dual-indexed cDNA libraries using the NEBNext Ultra II RNA Library Prep Kit for Illumina (E7770L, New England Biolabs). Ten cycles of PCR amplification were applied to all libraries. Sequencing-ready libraries were quantified using the Qubit dsDNA HS assay (Q32854, Invitrogen) and quality controlled with the Bioanalyzer 2100 DNA HS assay (5067–4626, Agilent). An equimolar pool was prepared with the 18 libraries for SE75 sequencing on a NextSeq550 device (Illumina). Sequencing output was 513.7 million 75-nt single-end reads and a minimum of 26 million reads were obtained for all samples. Resulting RNA-seq reads were processed and aligned using STAR[55] at the Biostatistics and Bioinformatics Unit, IRB Barcelona. Differential expression analysis was performed using DESeq2[56] and GSEA software[57] was used for enrichment analysis, including 10,000 permutations based on gene-set size. cGAS-STING and autophagy gene-sets were defined as previously described[58]. ISG analysis and interferon type classification were conducted using Interferome 2.0[20]. Heatmaps were generated using the *tidyverse*, *ggplot2*, *viridis*, *RColorBrewer* and *pheatmap* packages in R.

### Flow cytometry

Cells were washed with PBS, trypsinized for 2–3 min and collected by centrifugation at $1200 \times g$ for 5 min. Cells were resuspended in complete medium containing 100 nM MitoTracker Green (M7514, Invitrogen) and 5 μM MitoSOX Red (M36008, Invitrogen) and incubated for 15 min at 37 °C. DAPI (1 μg/μL) was added for dead cell exclusion and tubes were kept in ice until analysis. Samples were analyzed using a CytoFlex S (V4-B2-Y4-R3; Beckman Coulter) flow cytometer and at least 10,000 events per sample were collected.

## Mitochondrial respirometry using Seahorse XFe24

Cells were seeded in a Seahorse XF24 Cell Culture Microplate (100777-004, Agilent) in complete medium and incubated overnight at 37 °C under 5% $CO_2$. Culture medium was replaced with DMEM supplemented with 10% FBS, 1X Pen/Strep. Medium was replaced with Seahorse XF DMEM Assay Medium (103680-100, Agilent) supplemented with 2 mM Pyruvate (103578-100, Agilent) and 25 mM glucose (103577-100, Agilent). Oligomycin, FCCP and Rotenone+Antimycin A (1 μM) were injected sequentially and OCR was measured using the Seahorse XFe24 Analyzer (Agilent).

## Image analysis

All image processing and analyses were performed using Fiji (ImageJ).

**Mitophagy analysis with mito-QC and macroautophagy analysis.** Images were pre-processed by subtracting background with a rolling ball radius of 25 pixels and applying a Gaussian blur filter with a Σ radius of 1. The same threshold was applied to each image to obtain GFP$^+$ and mCherry$^+$ masks. Next, the GFP$^+$ mask was subtracted from the mCherry$^+$ mask to isolate the signal corresponding to mitolysosomes. Finally, individual mitolysosomes were quantified using 3D Objects Counter[59]. Volume of the GFP$^+$ mask is reported as mitochondrial mass. A similar approach was used to assess the number of LC3$^+$ autophagosomes, LC3$^+$GFP$^+$ mitophagosomes, LAMP1$^+$ lysosomes, and LC3$^+$LAMP1$^+$ autolysosomes.

**phospho-Ubiquitin$^{Ser65}$ levels.** Images were pre-processed by applying the *Despeckle* algorithm and subtracting background with a rolling ball radius of 25 pixels. The mean fluorescence intensity (cells) or thresholded pUb$^{Ser65+}$ area (tissue) was quantified in the maximal projection.

**Cytosolic DNA nucleoid quantification.** Images were pre-processed by subtracting background with a rolling ball radius of 25 pixels and applying a Gaussian blur filter with a Σ radius of 0.75. The same threshold was applied to each image to obtain TOMM20$^+$ and DAPI$^+$ masks. Cells were manually delineated and TOMM20$^+$ + DAPI$^+$ masks were subtracted from the DNA signal to isolate cytosolic DNA$^+$ foci. DNA$^+$ foci were quantified in each cell using the *Find Maxima* function. *Synaptic integrity assessment.* The number of CtBP2$^+$ terminals, mGluR6$^+$ terminals, and mGluR6$^+$CtbP2$^-$ faulty synapses were manually quantified using *Cell Counter*.

**4-Hydroxynonenal accumulation.** Images were pre-processed by subtracting background with a rolling ball radius of 15 pixels and applying a Gaussian blur filter with a Σ radius of 0.75. The same threshold was applied to each image to obtain 4-HNE$^+$ masks. 4-HNE$^+$ aggregates were quantified using 3D Objects Counter[59].

**DNA and DNA-bound cGAS quantification.** Images were pre-processed by subtracting background with a rolling ball radius of 20 pixels and applying a Gaussian blur filter with a Σ radius of 0.75. The same threshold was applied to each image to obtain cGAS$^+$ and DNA$^+$ masks. Colocalization analysis was performed using JACoP[59] and the overlap coefficient was determined.

**Cardiolipin and cardiolipin-TOMM20 colocalization quantification.** Mean fluorescence intensity of NAO was quantified in maximal projections. Colocalization analysis was performed using JACoP[59] and the overlap coefficient was determined.

**p62 and total ubiquitin levels analysis.** Mean fluorescence intensity was quantified in maximal projections.

**Retinal cell type quantification.** The number of Brn3a$^+$ RGCs, PKC$^+$ bipolar interneurons, ConeArrestin$^+$ cone photoreceptors, Iba1$^+$ microglia, and GFAP$^+$ astrocytic projections were manually quantified using *Cell Counter*. VisualArrestin$^+$ OS length was manually measured and internalization was reported as the ratio between OS and IS mean fluorescence intensity in maximal projections.

## Statistical analysis

Sample size for mouse experiments was determined based on previous experimental designs from the lab. Data were evaluated for normality and heteroscedasticity. Normally distributed data were analyzed using a one-way or two-way ANOVA with Šidák's or Tukey's post-hoc comparisons (more than two groups) or a two-tailed Student's *t* test (two groups). Non-normally distributed data were analyzed using the Mann–Whitney's *U*-test. Spearman's rank correlation $\rho$ coefficient ($r$) was used to assess the correlation between transcript levels and individual age. All statistical tests were performed with GraphPad Prism 9.0 and data were presented as the mean ± standard error of the mean (s.e.m).

## Reporting summary

Further information on research design is available in the Nature Portfolio Reporting Summary linked to this article.

## Data availability

Bulk RNA-seq sequencing data have been deposited in Gene Expression Omnibus (GEO) with accession number GSE231882. Fibroblast data: GSE113957 and C57BL/6J mice tissue data: GSE141252 are publicly available. Source data are provided with this paper.

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

## Acknowledgements

We thank Dr. Ian Ganley for generating and providing the *mito*-QC reporter mice; the Functional Genomics (Freddy Monteiro) and Bioinformatics (Camille Stephan-Otto Attolini) core facilities at IRB Barcelona for assistance with RNA-seq studies; the animal facility (Angélica Horrillo, María Tijero) at the CIB Margarita Salas; the animal cell culture facility (Zahira Corrales) and the light microscopy units at the CIB Margarita Salas (M. Teresa Seisdedos, Gema Elvira) and CNB (Ana Oña, Jaime Fernández); Cristina de Jesús Sen for assistance with mitochondria experiments and Jorge Montesinos for advice on subcellular fractionation. Diagrams were generated using BioRender. The authors thank Owen Howard for English-language editing and all other members of the Autophagy Lab for thoughtful discussions and support. Research in the P.B. lab is supported by grants 310030_215271 Swiss National Science Foundation (SNSF) and PID2021-126864NB I00 from MCIN, Spain. Research in J.C.E lab is supported by PID2019-103914RB-I00 from MICIN (MCIN/AEI/10.13039/501100011033, Spain, Unión Europea NextGenerationEU/PRTR). Both labs are also supported by CSIC Interdisciplinary Thematic Platform PTI-NEURO-AGING+ of the Spanish National Research Council (CSIC) and funded B.V.Z. Research in the AGD lab is supported by grants PID2020-114709RA-I00 and RYC2020-029291-I from MICIN and IDEAS222813GOME from AECC. J.I.J.-L. and J.Z.-M. are recipients of FPI and FPU predoctoral fellowships, respectively, from MCIN.

## Author contributions

Conceptualization: J.I.J.L., B.V.Z., A.G.D., P.B.; methodology: J.I.J.L., B.V.Z., A.V.P., J.Z.M., A.G.D., M.D.F.L., F.A.T.B., J.C.E.; investigation: J.I.J.L., B.V.Z., A.V.P., R.B.F., J.Z.M., M.D.F.L.; visualization: J.I.J.L., B.V.Z., A.V.P., J.Z.M.; funding acquisition: P.B.; project administration: P.B.; supervision: E.A.G., P.B.; writing – original draft: J.I.J.L., P.B.; writing – review & editing: J.I.J.L., B.V.Z., E.A.G., A.G.D., P.B. All authors read and approved the final version of the manuscript.

## Competing interests

The authors declare no competing interests.
