## [Peer Review File · Nature Communications]

Mitophagy curtails cytosolic mtDNA-dependent activation of cGAS/STING inflammation during agingREVIEWER COMMENTS

Reviewer #1 (Remarks to the Author):

The article by Jiménez-Loygorri et al demonstrates that mitophagy is increased with age in several organs, including the retina, in the mito QC reporter mouse. Moreover, in old animals, the cGAS/STING pathway and subsequently, type I IFN, are increased due to the elevation of cytoplasmic mtDNA. Activation of mitophagy in old mice rescued cGAS/STING activation and ameliorated deterioration of neurological function. The experiments are well thought out and the study is novel to the field's interest. A few concerns are as follows.

Major:

Fig 1: The authors show that that the age-associated increase in mitophagy is driven by the PINK1/Parkin pathway by looking into the levels of p-Ub (S65). What would be the influence of other receptor-mediated mitophagy proteins, such as FUNDC1, BNIP3/NIX, FKBP8, PHB2, and CL in the process?

Fig. 2: In Fig. 2A, the authors show several pathways which are upregulated in old mice by KEGG analysis and identified the IFNs to be the most highly enriched hallmarks. A detailed log fold change with p-value (with or without adjustment) must be added (perhaps in a table) to justify the choice of only looking into the IFNs.

Fig. 2D: The authors should specify the retinal region where they are seeing the cytoplasmic DNA by immunostaining. Counter-staining with specific antibodies or showing/labeling the layers of the retina is important.

Fig. 2F: The authors show the activation of the cGAS/STING pathway leads to the upregulation of IRF3 downstream genes. However, the pathway should be further assessed using STING inhibitor(s) or by knocking down the STING gene in vivo or at least in vitro.

Fig 3: As stated above, some evidence should be provided that indicates that other mitophagy pathway mediators are not involved in the process other than PINK1/Parkin-dependent pathway.

Figs. 4 and 5: The authors provide insights that activating mitophagy can ameliorate age-associated neurological changes with increased cGAS/STING response. However, the following aspects are not clear:

a) Since there is active mitophagy in older animals, the need to further activate mitophagy in these animals must be clarified- is the rate (flux) of mitophagy or autophagy low in older animals? Improvement of mitophagy upon UA treatment has been well-studied. The authors could use a mouse model with mitophagy/autophagy abnormalities, treat the animals with UA and then show rescue to further justify the claims made in this interesting manuscript.

b) Why is there an increased cGAS/STING response even though there is relatively elevated mitophagy in the retina? Are there other signaling pathways, particularly those involving lysosomes, which may be involved in cGAS/STING regulation during physiological aging? Or is the rate of mitophagy in normal aging not enough to limit cGAS/STING activation?

Some additional in-depth experiments must be performed to specifically answer these questions.

Minor:

All retinal cryo-sections must be labeled with the name of retinal layers for better readership.

Fig. 5D: The authors should counterstain GFAP with CRALBP to prove Müller cell activation. Also, images with higher/better resolution must be included.

Fig. 5E: The infiltration of the microglia is hard to visualize images with higher/better resolution and proper labeling (including arrows) should be included.

Page 3, Line 36: define PINK1

Page 4, Line 66: define SQSTM1

Page 5, Lines 80-81, 90: define proteins upon first use

Page 6, Line 106: define IRF3

Page 9, Line 192: define GFAP and Iba1

Results: City/state locations of companies are sometimes mentioned, but sometimes missing. Keep consistent throughout the entire section. Some catalog numbers are missing-please double check.

Page 13, Line 285: do not capitalize "western"

Page 14, Line 294: add the company to Ponceau S staining item (Sigma?)

Reviewer #2 (Remarks to the Author):

The authors suggest that mitophagy activity is high in the brain, retina, and kidney by using mito-QC mice. Paradoxically, in aged mice, abnormal mitochondrial morphology, an increase in cytosolic DNA, and upregulation of the cGAS-STING pathway are observed. The STING pathway is also activated in fibroblasts derived from elderly humans. Administration of urolithin A resulted in the activation of mitophagy, along with improvements in various aging phenotypes, cytosolic DNA accumulation, and STING pathway activation. Based on these observations, the authors suggest that the activation of mitophagy is effective for anti-aging.

Although this study encompasses the intriguing topics of mitophagy and aging, the presented data are not sufficient to support the authors' conclusions. Additionally, it is uncertain if this study has enough novelty.

Major points:

1. There are concerns regarding the interpretation of data suggesting mitophagy activation in aging mice using the mito-QC reporter (mCherry-GFP-MTS) mouse model. Given that it is reported that lysosomal activity decreases during the aging process in various organisms (PMID: 35741106, 8157122, 104881, 32482227, 23172144, 33446552), degradation of mCherry within lysosomes should also be delayed even if GFP is rapidly degraded. This would likely increase the number of mCherry-positive (and GFP-negative) structures (which the authors count as mitolysosomes) may increase. Comparing the number of mCherry structures between individuals with different lysosomal activity (i.e., young and old) would be thus difficult. In fact, this limitation has been shown in a previous paper (PMID: 32598458), which shows that RFP-single positive structures could accumulate upon lysosome inhibition and proposes that mitophagic flux should be determined in the absence and presence of lysosomal inhibitors. In the present study, it cannot be ruled out that mitophagy activity may rather be decreased; in fact, there is a high possibility of this.

2. Even if mitophagy is indeed activated in aged mice, its significance is unclear. Do the authors consider it to be a mere reactive response (but insufficient)? Is this the reason why there is no correlation between mitophagic activity and STING activation (Fig. 1 vs. Extended Data Fig. S3)?

3. The effect of urolithin A (UA) is questionable. Compared to young mice, the effect of UA on mitophagy in old mice is very slight (about 1.3 times?) (Fig. 4B, C). It is unlikely that such a small difference would affect the phenotype. Furthermore, the data merely show a correlation, and there is no evidence that UA exerts its effects through mitophagy. Additionally, the authors claim that the quality of mitochondria has been improved by UA administration based solely on the observation of 4-HNE, an oxidative stress marker (Fig. 4I), but this alone is insufficient evidence. More data (e.g., on mitochondrial activity and morphology) should also be presented.

4. The involvement of PINK1-Parkin is unclear. Since Parkin KO mice do not show obvious phenotypes unless they are not exercised, it is unlikely that Parkin is involved in natural aging phenotypes like those observed in this study.

5. Overall, the novelty of this paper is limited. Several papers have already suggested the relationship between cGAS-STING and aging, as well as the anti-aging effects of urolithin A.

Minor points:

1. As lipofuscin accumulation is prominent in aged mice (PMID: 36834842, 21804079), the authors should demonstrate that it did not affect the fluorescence imaging in this study.
2. Line 86: Cytochrome c is an intermembrane protein, not a matrix protein.
3. Fig. 1B: The mCherry puncta in the kidney appear to be autofluorescence near the brush border of the proximal tubule rather than intracellular. Clearer, magnified images should be provided as insets.
4. Several figures lack scale bars (Fig. 1B, 5D, E, etc.).
5. Lines 54 and 56: The detection of the mito-QC reporter is not by "immunofluorescence".

Reviewer #3 (Remarks to the Author):

This study aimed to investigate the role of mitophagy in regulating systemic inflammation during aging regulated by cytosolic mtDNA/cGAS/STING signaling pathway. This study is well designed and holds promise and significance. However, the novelty of the study is limited and there are several issues that need to be further addressed.

Major concerns

1. The significance of mitophagy in mitochondrial quality control, mitochondrial damage associated with aging, activation of mtDNA for STING, and STING signaling triggered by cytosolic mtDNA leakage has been previously reported. Therefore, the novelty and originality of this research is limited to a certain extent. This paper primarily validates previously published findings in multiple organs and tissues, however, without adding significant underlying novel mechanisms.
2. The author observed enhanced mitophagy activation in multiple organs of aged mice, and drug-induced mitophagy alleviated cytosolic mtDNA leakage and STING activation. However, it remains unclear why aged mouse organs exhibit increased mitophagy activation accompanied by elevated STING activation. Is the physiological activation of mitophagy inadequate to clear damaged mitochondria or mtDNA in aged mice? Furthermore, Young mouse organs exhibited weaker mitophagy and STING

activation. Aging-associated mitochondrial damage and STING activation may hold greater scientific significance but is unexplored in this paper.

3. The author mainly utilized mito-QC mice to assess mitophagy at various tissue levels. More methods/assays to detect mitophagy and related results should be provided to further solidify the findings.

4. Defective autophagy/mitophagy in aged cells/tissues has been reported, the author should discuss about the inconsistency.

5. To confirm the activation of STING signaling by cytosolic mtDNA, the authors may provide the experiment to test the role of cytosolic mtDNA scavenge in modulating STING activation in old cells/tissues.

Minor concerns

1. As shown in the extended data graph. S2F, in aged mice, electron microscopy showed an increase in the amount of mitochondrial swelling and increased disruption of the inner and outer mitochondrial membranes, but not in young mice. Since mitochondrial damage is an inducer of mitophagy, there is a need to increase the relevant experiments comparing mitophagic activation in young and old mice under stressful conditions.

2. The author used UA treatment to induce mitophagy in young and aged mice for 8 weeks, what the effect of mitophagy induction on the tissues in young mice?

Reviewer #1 (Remarks to the Author):

The article by Jiménez-Loygorri et al demonstrates that mitophagy is increased with age in several organs, including the retina, in the mito QC reporter mouse. Moreover, in old animals, the cGAS/STING pathway and subsequently, type I IFN, are increased due to the elevation of cytoplasmic mtDNA. Activation of mitophagy in old mice rescued cGAS/STING activation and ameliorated deterioration of neurological function. The experiments are well thought out and the study is novel to the field's interest. A few concerns are as follows.

We thank reviewer #1 for their positive feedback and the constructive criticism that has helped strengthen the findings of our paper.

Major:

Fig 1: The authors show that that the age-associated increase in mitophagy is driven by the PINK1/Parkin pathway by looking into the levels of p-Ub (S65). What would be the influence of other receptor-mediated mitophagy proteins, such as FUNDC1, BNIP3/NIX, FKBP8, PHB2, and CL in the process?

Thanks for this insightful suggestion. Even though we observed a clear increase in PINK1/Parkin-activity evidenced by high pUb^{Ser65} levels (**Fig. 1d**), it cannot be discarded that other mitophagy pathways could be simultaneously upregulated in parallel. To address this concern, we have measured the protein levels of ubiquitin-independent mitophagy receptors BNIP3L/NIX, BNIP3, FKBP8, PHB2 and FUNDC1, in the retina of young and old mice. We validated that there were no significant changes in any of them (**Supplementary Fig. 2b**).

Another alternative pathway is lipid-mediated mitophagy, that is driven by changes in mitochondria lipid composition, such as cardiolipin translocation from the inner to the outer mitochondrial membrane¹. We measured total cardiolipin (CL) levels using the probe Nonyl acridine orange (NAO) and, even though we found a significant reduction in old retinas, there were no changes in colocalization with the mitochondrial outer membrane protein TOMM20 (**Supplementary Fig. 2c**). This total CL decrease might represent an unrelated lipidome remodeling in the aging retina, notably photoreceptors are particularly enriched in cardiolipin and its function and number decrease with age².

These new data support that age-associated mitophagy increase is not due to upregulation of receptor-mediated or lipid-mediated pathways.

Fig. 2: In Fig. 2A, the authors show several pathways which are upregulated in old mice by KEGG analysis and identified the IFNs to be the most highly enriched hallmarks. A detailed log fold change with p-value (with or without adjustment) must be added (perhaps in a table) to justify the choice of only looking into the IFNs.

We have now included a full list of Hallmark and KEGG enrichment analysis results (NES, *p*-value, *q*-value) in the **Source Data** provided together with the manuscript. As shown in the complete table, IFN- α response, IFN- γ response and TNF- α signaling via NF- κ B were the top three Hallmark gene sets enriched in the old retina (**Fig. 2b**) and the top three gene sets downregulated with UA treatment (**Fig. 5b**).

Fig. 2D: The authors should specify the retinal region where they are seeing the cytoplasmic DNA by immunostaining. Counter-staining with specific antibodies or showing/labeling the layers of the retina is important.

Thanks for this suggestion that will ease the comprehension of our results, we have now labeled neuroretina layers in all pertinent figures (Figs. 1, 2, 4, 5; Supplementary Figs. 1, 2, 3, 8). In Fig. 2d, the outer plexiform layer (OPL) and outer nuclear layer (ONL) are shown.

Fig. 2F: The authors show the activation of the cGAS/STING pathway leads to the upregulation of IRF3 downstream genes. However, the pathway should be further assessed using STING inhibitor(s) or by knocking down the STING gene *in vivo* or at least *in vitro*.

We agree with the reviewer on this point and acknowledge that conducting the same analyses in STING-deficient mice expressing the *mito*-QC reporter would strengthen our conclusions. However, breeding and aging (>22 months) a big enough cohort would take at least three years to complete and is not feasible. Furthermore, even though murine STING inhibitors³ are commercially available there is no evidence on whether they are able to cross the blood-retina or blood-brain barrier.

To address this issue, we have set up an *in vitro* model of mtDNA release using Bcl-2 inhibitor ABT-737 (**Fig. 6b**) that recapitulates our findings *in vivo*. Bcl-2 inhibition promotes Bax/Bak activation, pore formation and mitochondrial outer membrane permeabilization, ultimately leading to inner membrane extrusion and release of matrix components including mtDNA. In a similar fashion to our findings in mice and primary fibroblasts, ABT-737-treated cells show increased cytosolic DNA (**Fig. 6c**) and activation of the cGAS/STING pathway (**Fig. 6d**). These changes were again amenable to downregulation by treatment with the mitophagy inducer UA and we also demonstrated that mitophagy is induced as a cytoprotective response as it can be shut off by co-treatment with cGAS inhibitor G140 (**Fig. 6e**).

Fig 3: As stated above, some evidence should be provided that indicates that other mitophagy pathway mediators are not involved in the process other than PINK1/Parkin-dependent pathway.

We have now performed a more detailed analysis of alternative mitophagy pathways in primary NHDFs from young and old donors (**Supplementary Fig. 6**). No significant changes were observed in the protein levels of mitophagy receptors BNIP3L/NIX, BNIP3, FKBP8, PHB2 or FUNDC1 (**Supplementary Fig. 6a**). Similarly, no differences were observed in either total CL levels or translocation to the outer mitochondrial membrane (**Supplementary Fig. 6b**). Thus, receptor-mediated and lipid-mediated mitophagy seem to be dispensable in age-associated mitophagy upregulation. In a similar fashion to the retina, PINK1/Parkin-dependent mitophagy seems to be the sole responsible for mitochondria quality control upon aging.

Figs. 4 and 5: The authors provide insights that activating mitophagy can ameliorate age-associated neurological changes with increased cGAS/STING response. However, the following aspects are not clear:

a) Since there is active mitophagy in older animals, the need to further activate mitophagy in these animals must be clarified- is the rate (flux) of mitophagy or autophagy low in older animals? Improvement of mitophagy upon UA treatment has been well-studied. The authors could use a mouse model with mitophagy/autophagy abnormalities, treat the animals with UA and then show rescue to further justify the claims made in this interesting manuscript.

We thank the reviewer for raising this point and agree that performing a similar intervention study in a mitophagy-deficient strain (*PINK1*^{-/-} or *Parkin*^{-/-}) would strengthen our findings, but as stated above crossing and breeding could take up to three years and is out of the scope of the current manuscript. Nonetheless, previous results from our lab have shown that *Ambra1*^{+/*gt*} mice, which have a slight autophagy reduction, age prematurely and present profound age-associated metabolic alterations concomitant with dysfunctional mitochondria accumulation⁴. AMBRA1 has also been recently described to be essential for PINK1 stabilization and downstream PINK1/Parkin-dependent mitophagy activation⁵. This evidence supports our hypothesis and again highlights the relevance of the PINK1/Parkin pathway in sustaining mitochondrial homeostasis upon aging.

Regarding mitophagy flux alterations upon aging, since age-associated mitophagy upregulation is one of the core findings of our manuscript, we have now performed additional proof-of-concept experiments to validate our findings using the *mito*-QC reporter. We injected young and old mice with the protease inhibitor Leupeptin and found that mitophagy flux (number of mitolysosomes with

Leupeptin/Vehicle) is indeed increased in the retina of old mice (2.5-fold change) compared to young mice (1.8-fold change) (**Extended Data for Reviewers 1a**). We have also compared the efficiency of autophagic cargo delivery to the lysosome and found that mitophagy efficiency (mitolysosome/mitophagosome ratio) was again higher in old mice (**Extended Data for Reviewers 1b**) while no apparent changes were observed in unspecific autophagy efficiency (autolysosome/autophagosome ratio) (**Extended Data for Reviewers 1b**).

Figure 1. Mitophagy flux changes upon aging. **a** Young (6 months) and old (26 months) *mito*-QC mice were injected intraperitoneally with 40 mg/kg Leupeptin or vehicle and sacrificed after 16 hours. Representative images and corresponding quantification of mitolysosome number (mCherry⁺GFP⁺ puncta). **b** Quantification of mitophagy efficiency (mitolysosomes per mitophagosome) from Supplementary Fig. 3a. **c** Quantification of autophagy efficiency (autolysosomes per autophagosome) from Supplementary Fig. 3a. Scale bar, 50 μ m (**a**). All data are expressed as the mean \pm s.e.m. Dots represent individual mice. P values were calculated using a 2-way ANOVA with Šídák's post-hoc test (**a**) or a two-tailed Student's *t*-test (**b**, **c**).

b) Why is there an increased cGAS/STING response even though there is relatively elevated mitophagy in the retina? Are there other signaling pathways, particularly those involving lysosomes, which may be involved in cGAS/STING regulation during physiological aging? Or is the rate of mitophagy in normal aging not enough to limit cGAS/STING activation?

In order to address why the age-associated mitophagy increase is not enough to limit cGAS/STING activation, we have now performed additional *in vivo* and *in vitro* analyses focused on mitochondria quality and quantity (**Fig. 6**). While we observed increased mitophagy in the old retina (**Fig. 1b**), there were no changes on mitochondrial mass (**Supplementary Fig. 4e**) and we also observed altered morphology (**Supplementary Fig. 4f**). Mitochondrial homeostasis requires a tight balance between degradation and biogenesis to maintain a healthy mitochondria pool that can cope with energetic demands ⁶.

Age-associated mitophagy could therefore aid in removing the population of leaky mitochondria that are releasing mtDNA to the cytosol, but proper mitochondrial biogenesis would be required to replace them and curtail this cGAS/STING upstream triggering stimulus. Since UA has been described to induce mitochondrial biogenesis ⁷, we assessed mitochondrial mass levels in our *in vivo* aging (**Fig. 6a**) and *in vitro* ABT-737 models (**Fig. 6f**, left) and found that UA also stimulated mitochondrial biogenesis in our experimental context. We had previously observed a reduction in age-related oxidative stress with UA (**Fig. 4i**), and we have now described a similar but more nuanced paradigm in the ABT-737 model using a mitochondrial O₂⁻ probe (**Fig. 6f**, right). Most importantly, mitochondrial respirometry shows that UA increases the spare respiratory capacity of cells and also restored basal respiration and ATP production in ABT-737-challenged cells (**Fig. 6g**). In summary, UA induced biogenesis and improved mitochondrial function. Simultaneous stimulation of mitophagy and mitochondrial biogenesis could be required to reduce cGAS/STING activation upon aging, we have elaborated on this concept in the extended discussion of the manuscript.

Finally, we cannot rule out that another age-related changes in the endolysosomal system or other autophagy pathways might also play a role in sustaining cGAS/STING activation. While we did not

observe any changes in the morphology or total number of lysosomes (**Supplementary Fig. 3a, b**), there was an accumulation of autophagosomes (**Supplementary Fig. 3b**) and traditional non-specific autophagic cargos (**Supplementary Fig. 3c, d**) in line with our previous findings showing impaired macroautophagy flux in the aged retina⁸. In steady-state conditions, cGAS/STING response is tuned down through autophagy-mediated STING degradation that is mediated by ULK1 activation due to increased levels of cytosolic cGAMP⁹. Furthermore, STING recognition requires the autophagy adaptor p62 that is also activated by TBK1¹⁰. In our experimental conditions, we indeed observed a drastic increase of STING protein levels (**Fig. 2f**) that was concomitant with increased number of p62⁺ aggregates (**Supplementary Fig. 3c**). This inability to switch off STING downstream signaling could also be contributing and exacerbating the inflammatory phenotype observed upon aging. Reduction of mtDNA leakage after treatment with UA would then be able to circumvent general autophagic flux deficiency, STING accumulation and, ultimately, reduce cGAS/STING-mediated inflammation.

Some additional in-depth experiments must be performed to specifically answer these questions.

Minor:

All retinal cryo-sections must be labeled with the name of retinal layers for better readership.

Retinal layers are now labeled in all pertinent figures (Figs. 1, 2, 4, 5; Supplementary Figs. 1, 2, 3, 8).

Fig. 5D: The authors should counterstain GFAP with CRALBP to prove Müller cell activation. Also, images with higher/better resolution must be included.

We thank the reviewer for this suggestion. We have performed CRALBP immunostaining successfully in the past but, after several tests with different antigen retrieval and permeabilization methods, have not been able to get specific signal with our current antibody lot (Abcam AB15051, lot 1012141-3). As an alternative, we have used anti-Glutamine Synthetase (GS) and replace the images in **Fig. 5d**.

In the final merged document including the main text and figures, resolution was altered after conversion to .png. We apologize for this issue and will include high-resolution images in the merged document as well as an additional .zip file with all figures.

Fig. 5E: The infiltration of the microglia is hard to visualize images with higher/better resolution and proper labeling (including arrows) should be included.

We have added arrowheads to **Fig. 5e**.

Page 3, Line 36: define PINK1

All protein names have now been defined upon first use.

Page 4, Line 66: define SQSTM1

Page 5, Lines 80-81, 90: define proteins upon first use

Page 6, Line 106: define IRF3

Page 9, Line 192: define GFAP and Iba1

Results: City/state locations of companies are sometimes mentioned, but sometimes missing. Keep consistent throughout the entire section. Some catalog numbers are missing-please double check.

We have now gone over the **Materials and methods** section of the paper, added missing information and unified nomenclature.

Page 13, Line 285: do not capitalize “western”

This has been corrected in the updated version of the manuscript.

Page 14, Line 294: add the company to Ponceau S staining item (Sigma?)

This has been corrected in the updated version of the manuscript.

Reviewer #2 (Remarks to the Author):

The authors suggest that mitophagy activity is high in the brain, retina, and kidney by using mito-QC mice. Paradoxically, in aged mice, abnormal mitochondrial morphology, an increase in cytosolic DNA, and upregulation of the cGAS-STING pathway are observed. The STING pathway is also activated in fibroblasts derived from elderly humans. Administration of urolithin A resulted in the activation of mitophagy, along with improvements in various aging phenotypes, cytosolic DNA accumulation, and STING pathway activation. Based on these observations, the authors suggest that the activation of mitophagy is effective for anti-aging.

Although this study encompasses the intriguing topics of mitophagy and aging, the presented data are not sufficient to support the authors' conclusions. Additionally, it is uncertain if this study has enough novelty.

We thank reviewer #2 for their valuable suggestions that have helped us to improve the quality of the revised manuscript.

Major points:

1. There are concerns regarding the interpretation of data suggesting mitophagy activation in aging mice using the mito-QC reporter (mCherry-GFP-MTS) mouse model. Given that it is reported that lysosomal activity decreases during the aging process in various organisms (PMID: 35741106, 8157122, 104881, 32482227, 23172144, 33446552), degradation of mCherry within lysosomes should also be delayed even if GFP is rapidly degraded. This would likely increase the number of mCherry-positive (and GFP-negative) structures (which the authors count as mitolysosomes) may increase. Comparing the number of mCherry structures between individuals with different lysosomal activity (i.e., young and old) would be thus difficult. In fact, this limitation has been shown in a previous paper (PMID: 32598458), which shows that RFP-single positive structures could accumulate upon lysosome inhibition and proposes that mitophagic flux should be determined in the absence and presence of lysosomal inhibitors. In the present study, it cannot be ruled out that mitophagy activity may rather be decreased; in fact, there is a high possibility of this.

We thank the reviewer for this suggestion and raising this very important point. In order to address this concern and those of reviewer #1, we have now run additional experiments to validate the increase in mitophagic flux observed in aged *mito-QC* mice. We performed the suggested experiment using intraperitoneal administration of the lysosomal degradation inhibitor Leupeptin (40mg/kg), which is able to cross the blood-retina barrier as previously reported by our group¹¹. When comparing Vehicle vs Leupeptin-treated mice, we observed increased mitolysosome accumulation in the retina of old mice (2.5-fold change) compared to young mice (1.8-fold change) (**Extended Data for Reviewers 1a**). This new data validates our findings (**Fig. 1b**) and supports an age-associated increase of mitophagy in the retina.

2. Even if mitophagy is indeed activated in aged mice, its significance is unclear. Do the authors consider it to be a mere reactive response (but insufficient)? Is this the reason why there is no correlation between mitophagic activity and STING activation (Fig. 1 vs. Extended Data Fig. S3)?

A similar point was raised by reviewer #1 (see response to point "Figs. 4 and 5"). We now have performed additional experiments using an *in vitro* model of mtDNA release and cGAS/STING activation in ARPE-19 cells treated with the Bcl-2 inhibitor ABT-737 (**Fig. 6b**). We show that mitophagy upregulation is, as suggested, a reactive response to cGAS/STING activation since it was inhibited by co-treatment with human cGAS inhibitor G140 (**Fig. 6e**). We hypothesize that while getting rid of faulty mitochondria might aid in reducing cGAS/STING activation, mitochondrial

biogenesis and improved mitochondrial function might also be necessary to reduce its triggering event: mtDNA leakage (see response to point 3).

3. The effect of urolithin A (UA) is questionable. Compared to young mice, the effect of UA on mitophagy in old mice is very slight (about 1.3 times?) (Fig. 4B, C). It is unlikely that such a small difference would affect the phenotype. Furthermore, the data merely show a correlation, and there is no evidence that UA exerts its effects through mitophagy. Additionally, the authors claim that the quality of mitochondria has been improved by UA administration based solely on the observation of 4-HNE, an oxidative stress marker (Fig. 4I), but this alone is insufficient evidence. More data (e.g., on mitochondrial activity and morphology) should also be presented.

A similar point was raised by reviewer #1 (see response to point “Figs. 4 and 5”). Generating Parkin- or PINK1-deficient mice would be ideal and clarify whether the beneficial effect of UA is solely due to mitophagy, however breeding and aging a colony of mitophagy deficient *mito-QC* mice would take up to three years and is out of the scope of the paper. Similarly, as of right now there are no specific mitophagy inhibitors commercially available and genetic approaches (*PINK1* or *PARK2* siRNA-mediated knockdown) have been described to stimulate type-I IFN inflammatory response by themselves¹² leading to confounding data interpretation.

We have now provided additional evidence proving improved mitochondrial homeostasis after UA treatment. Notably, UA induced mitochondrial biogenesis in old mice (Fig. 6a) and cells treated with ABT-737 (Fig. 6f, left). We also observed a tendency towards decreased CI- and CIII-associated mitochondrial superoxide production in ABT-737-challenged cells (Fig. 6f, right). In accordance, UA improved mitochondrial spare respiratory capacity, its ability to cope with increased energy demands, and also restored basal respiration and ATP production in cells treated with ABT-737 (Fig. 6g). We have elaborated on this idea in the now-extended discussion of the manuscript.

4. The involvement of PINK1-Parkin is unclear. Since Parkin KO mice do not show obvious phenotypes unless they are not exercised, it is unlikely that Parkin is involved in natural aging phenotypes like those observed in this study.

The reviewer raises an important point here, it has been extensively described that *PINK1*^{-/-} and *Parkin*^{-/-} mice do not show any major phenotypic or behavioral alterations. However, a study has shown that PINK1-deficient mice do show mitochondrial function alterations that are further exacerbated in aged animals¹³. To our knowledge, there are no studies focused on neuroinflammation or mitophagy in aged PINK1 or Parkin-deficient mice. A recent manuscript has also demonstrated that loss of STING improves the mitochondrial and motor phenotype observed in *parkin*^{-/-} *Drosophila melanogaster*¹⁴. It would be interesting to investigate if this is also the case in mammals, and particularly in mice regarding the mitochondrial phenotype, but we believe it is out of the scope of the present manuscript.

5. Overall, the novelty of this paper is limited. Several papers have already suggested the relationship between cGAS-STING and aging, as well as the anti-aging effects of urolithin A.

While it is true that the geroprotective effect of UA and cGAS-STING response during aging have been widely studied, herein we have been able to link both and also provide novel insight:

1. We have performed the first organism-wide systematic analysis of mitophagy changes upon aging in mice, using a validated and uniform methodology (Fig. 1b).
2. While PINK1/Parkin-dependent mitophagy has been described to be dispensable in maintaining basal mitochondrial quality control in young mice¹⁵, here we find that it is selectively upregulated upon aging (Fig. 1d). The pathways that have been traditionally associated with basal mitophagy are not involved in age-associated mitophagy upregulation (Supplementary Fig. 1b, c).
3. Previously, it had been described that mice with deficient PINK1/Parkin-mediated mitophagy present exacerbated inflammatory response to exhaustive exercise¹⁶. However, this is the

first time where the opposite strategy (boosting mitophagy) has been used to limit cGAS/STING-mediated activation (**Fig. 7**).

4. We propose for the first time cytosolic DNA sensing by cGAS as a direct inducer of mitophagy (**Fig. 6e**).
5. We also show that UA manages to slow down age-related retinal degeneration, most importantly it helped preserve synaptic integrity (**Fig. 4h**) and visual function (**Fig. 4g**).
6. Other studies have focused on cGAS/STING inhibition to reduce aging-associated inflammation¹⁷. While this strategy efficiently decreases inflammation, the side-effects of prolonged shut down of an immune pathway are unclear. This approach could hardly be translated to the clinic or be applicable in the context of aging. In our study, we indirectly reduce inflammation by promoting mitochondrial homeostasis using UA, a readily available natural compound, and have discussed this aspect of our work in the now-extended discussion.

Taking all of this into account, we believe that our work provides a useful resource for the scientific community, presents enough novelty and helps tackle age-associated inflammation from a new angle.

Minor points:

1. As lipofuscin accumulation is prominent in aged mice (PMID: 36834842, 21804079), the authors should demonstrate that it did not affect the fluorescence imaging in this study.

We thank the reviewer for this suggestion. We have performed a λ -scan autofluorescence analysis using $\lambda_{ex} = 405$ nm and observed increased levels of lipofuscin ($\lambda_{em} = \sim 500$ nm) in the retinas of old mice (**Extended Data for Reviewers 2a, b**) as previously reported in the literature⁸. However, we observed no changes in the colocalization of mCherry and Lipofuscin with age (**Extended Data for Reviewers 2c**), indicating that it does not interfere with *mito*-QC signal and quantification.

These interesting data, together with our report that mitophagy and general macroautophagy present divergent regulation upon aging (**Fig. 1b, Supplementary Fig. 3a, b**), indicates that there might be a specific subset of lysosomes in charge of mitophagic degradation that do not present as much age-associated lipofuscin deposition. We had previously reported that chaperone-mediated autophagy (CMA), a process that requires a specific subset of LAMP2A⁺Hsc70⁺ lysosomes, is also upregulated in the aging retina to counteract deficient macroautophagy^{4, 8}). Further studies are needed to fully elucidate lysosomal population heterogeneity, specifically in the context of the retina.

Figure 2. Lipofuscin does not interfere with *mito*-QC. **a** λ -scan analysis ($\lambda_{ex} = 405$ nm) in whole-eye cryosections of young (8 months) and old (24 months) *mito*-QC reporter mice. **b** Representative images showing *mito*-QC reporter and lipofuscin ($\lambda_{em} = 500$ nm; cyan) signal. **c** Quantification of colocalization (Pearson's Coefficient) between mCherry and lipofuscin ($\lambda_{em} = 500$ nm). Scale bar, 50 μ m (**b**). All data are expressed as the mean \pm s.e.m. Dots represent individual mice. *P* values were calculated using a two-tailed Student's *t*-test (**c**).

2. Line 86: Cytochrome c is an intermembrane protein, not a matrix protein.

This has now been corrected in the updated manuscript.

3. Fig. 1B: The mCherry puncta in the kidney appear to be autofluorescence near the brush border of the proximal tubule rather than intraacellular. Clearer, magnified images should be provided as insets.

We have now included a new figure magnified images for some of the organs where mitophagy was increased upon aging (**Supplementary Fig. 1**). Regarding our results on kidney, thanks to the development of *mito-QC* it has been shown that it is one of the organs with the highest levels of mitophagy, particularly prominent in tubules of the renal cortex where it colocalized with lysosomal marker LAMP1¹⁸. Traditional EM shows that the apical region right next to the brush border is where most lysosomes are located while mitochondria localize to the basal region¹⁹. It has also been described that mitophagy in the proximal tubule plays an important role in the response to stress conditions, such as ischemia-reperfusion injury²⁰, and this could also be the case during physiological aging that involves progressive kidney dysfunction²¹.

4. Several figures lack scale bars (Fig. 1B, 5D, E, etc.).

We apologize for the inconvenience; clearly visible scale bars are now included in all figures.

5. Lines 54 and 56: The detection of the mito-QC reporter is not by "immunofluorescence".

This has now been corrected in the updated manuscript.

Reviewer #3 (Remarks to the Author):

This study aimed to investigate the role of mitophagy in regulating systemic inflammation during aging regulated by cytosolic mtDNA/cGAS/STING signaling pathway. This study is well designed and holds promise and significance. However, the novelty of the study is limited and there are several issues that need to be further addressed.

We thank the reviewer for the positive comments and considering our work valuable for the field.

Major concerns

1. The significance of mitophagy in mitochondrial quality control, mitochondrial damage associated with aging, activation of mtDNA for STING, and STING signaling triggered by cytosolic mtDNA leakage has been previously reported. Therefore, the novelty and originality of this research is limited to a certain extent. This paper primarily validates previously published findings in multiple organs and tissues, however, without adding significant underlying novel mechanisms.

This point has also been raised by reviewer #2 (see response to Point 5). While it is true that we have had to validate previous findings in our experimental setting, we provide novel insight on the interplay between mitophagy and inflammation as well as describe a new aspect to the geroprotective effect of UA.

2. The author observed enhanced mitophagy activation in multiple organs of aged mice, and drug-induced mitophagy alleviated cytosolic mtDNA leakage and STING activation. However, it remains unclear why aged mouse organs exhibit increased mitophagy activation accompanied by elevated STING activation. Is the physiological activation of mitophagy inadequate to clear damaged mitochondria or mtDNA in aged mice? Furthermore, Young mouse organs exhibited weaker

mitophagy and STING activation. Aging-associated mitochondrial damage and STING activation may hold greater scientific significance but is unexplored in this paper.

We acknowledge that the correlation between mitophagy levels and STING activation might be perplexing, and have now explored how UA-induced mitochondrial biogenesis could also be contributing to reducing mtDNA leakage (**Fig. 6a, c, f**) and improving overall mitochondrial function (**Fig. 6f, g**).

Regarding mitochondrial damage and STING activation, new additional experiments show that mitophagy is directly activated by cGAS/STING signaling, as it was abolished by cGAS inhibitor G140 (**Fig. 6d, e**) even if mtDNA release was as the same level cells treated only with ABT-737 (**Fig. 6c**). Please, see response to reviewer #1 point "Figs. 4 and 5" and reviewer #2 points "2" and "3".

3. The author mainly utilized mito-QC mice to assess mitophagy at various tissue levels. More methods/assays to detect mitophagy and related results should be provided to further solidify the findings.

We agree with the reviewer on the use of orthogonal methodologies being essential to validate our claims. Apart from the *mito*-QC reporter (**Fig. 1b**), in the original paper we also provided immunofluorescence staining against pUbiquitin^{Ser65} indicating PINK1/Parkin pathway activity (**Fig. 1c, d, Supplementary Fig. 2a**). We have now performed additional western blotting to assess the protein levels of mitophagy receptors (**Supplementary Fig. 2b**) and mitochondrial cardiolipin labeling (**Supplementary Fig. 2c**), to discard the involvement of alternative mitophagy pathways. We conclude that age-associated mitophagy upregulation is solely due to increased PINK1/Parkin-dependent mitophagy.

To validate our findings on mitophagy upregulation we have also measured mitophagic flux in mice using lysosomal inhibitors (**Extended Data for Reviewers 1a**) and mitophagy efficiency by comparing mitolysosome and mitophagosome number (**Supplementary Fig. 3a, Extended Data for Reviewers 1b**) (Please see response to point reviewer #1 point "Figs. 4 and 5"). All this evidence supports a robust mitophagy increase upon aging in the retina.

4. Defective autophagy/mitophagy in aged cells/tissues has been reported, the author should discuss about the inconsistency.

We thank the reviewer for raising this timely question. As mentioned in the introduction of the manuscript, there are several conflicting reports in the literature regarding mitophagy levels and aging. For example, there is a study suggesting skeletal muscle presents upregulated mitophagy in old mice²² while another one indicates that mitophagy is downregulated²³, both drawing these opposite conclusions from the same results: decreased PINK1 protein levels measured by immunoblotting. Mitophagy is a highly dynamic process and therefore it is complicated to make a definitive statement using traditional techniques, in the example above decreased PINK1 levels could be interpreted either as increased degradation (mitophagy upregulation) or decreased stabilization (mitophagy downregulation). Similarly, traditional autophagy flux (+lysosomal inhibitors) assessment by western blot is not reliable to measure changes in mitophagy since the fraction of mitochondria that undergo mitophagy is very small compared to total mitochondrial mass.

Thanks to the development of specific tandem fluorescence reporters, we are now able to monitor mitophagy by measuring its endpoint, mitochondrial delivery to the lysosome²⁴. In the present manuscript we performed a whole-body analysis using this methodology and found that mitophagy either remains stable or increases with age (**Fig. 1**). The same phenomenon has been described in the model organism *Drosophila melanogaster* using the mtKeima tandem reporter²⁵.

5. To confirm the activation of STING signaling by cytosolic mtDNA, the authors may provide the experiment to test the role of cytosolic mtDNA scavenge in modulating STING activation in old cells/tissues.

To address this point, we have performed additional *in vitro* experiments using a model of mtDNA release (Fig. 6b). In ABT-737-treated cells, even though cytosolic DNA levels remain the same (Fig. 6c), STING stabilization was reduced when we inhibited upstream cGAS using G140 (Fig. 6d). Therefore, we have validated that mtDNA sensing by cGAS induces STING activation.

Minor concerns

1. As shown in the extended data graph. S2F, in aged mice, electron microscopy showed an increase in the amount of mitochondrial swelling and increased disruption of the inner and outer mitochondrial membranes, but not in young mice. Since mitochondrial damage is an inducer of mitophagy, there is a need to increase the relevant experiments comparing mitophagic activation in young and old mice under stressful conditions.

We have subjected young and old mice to traditional starvation to check if mitophagy could still be further upregulated in response to a stressful condition that involves increased nutrient demand. As shown in **Extended Data for Reviewers 3**, we observed a similar increase in mitophagy levels when comparing young (1.8-fold change) and old mice (1.9-fold change) in starvation or fed *ad libitum*. This new data, together with our new results on mitophagic flux with lysosomal inhibitors (**Extended Data for Reviewers 1a**), suggests that mitophagy is still fully functional in old mice.

Figure 3. Mitophagy is upregulated during starvation. Young (6 months) and old (26 months) *mito-QC* mice were subjected to two cycles of 16 hour starvation. Representative images and corresponding quantification of mitolysosome number. Scale bar, 50 μ m. All data are expressed as the mean \pm s.e.m. Dots represent individual mice. *P* values were calculated using a 2-way ANOVA with Šidák's post-hoc test.

2. The author used UA treatment to induce mitophagy in young and aged mice for 8 weeks, what the effect of mitophagy induction on the tissues in young mice?

While we observed a very robust increase on mitophagy levels in young mice (Fig. 4b) and some transcriptomic changes (Supplementary Fig. 9), there were no differences in any of the histological, electrophysiological or behavioural parameters analyzed (Fig. 4d-j, 5a-e, Supplementary Fig. 8). This is in line with the literature, that has described that UA is able to modulate and improve mitochondrial parameters without producing any apparent side-effects in middle-aged²⁶ and elderly^{27,28} individuals.

References

1. Teresak P, Lapao A, Subic N, Boya P, Elazar Z, Simonsen A. Regulation of PRKN-independent mitophagy. *Autophagy* **18**, 24-39 (2022).
2. Ferdous S, *et al.* Age-Related Retinal Changes in Wild-Type C57BL/6J Mice Between 2 and 32 Months. *Invest Ophthalmol Vis Sci* **62**, 9 (2021).

3. Haag SM, *et al.* Targeting STING with covalent small-molecule inhibitors. *Nature* **559**, 269-273 (2018).
4. Ramírez-Pardo I, *et al.* Ambra1 haploinsufficiency in CD1 mice results in metabolic alterations and exacerbates age-associated retinal degeneration. *Autophagy*, 1-21 (2022).
5. Di Rienzo M, *et al.* AMBRA1 regulates mitophagy by interacting with ATAD3A and promoting PINK1 stability. *Autophagy* **18**, 1752-1762 (2022).
6. Palikaras K, Lionaki E, Tavernarakis N. Coordination of mitophagy and mitochondrial biogenesis during ageing in *C. elegans*. *Nature* **521**, 525-528 (2015).
7. Ryu D, *et al.* Urolithin A induces mitophagy and prolongs lifespan in *C. elegans* and increases muscle function in rodents. *Nat Med* **22**, 879-888 (2016).
8. Rodriguez-Muela N, *et al.* Balance between autophagic pathways preserves retinal homeostasis. *Aging cell* **12**, 478-488 (2013).
9. Konno H, Konno K, Barber GN. Cyclic dinucleotides trigger ULK1 (ATG1) phosphorylation of STING to prevent sustained innate immune signaling. *Cell* **155**, 688-698 (2013).
10. Prabakaran T, *et al.* Attenuation of cGAS-STING signaling is mediated by a p62/SQSTM1-dependent autophagy pathway activated by TBK1. *Embo j* **37**, (2018).
11. Esteban-Martinez L, Boya P. Autophagic flux determination in vivo and ex vivo. *Methods* **75**, 79-86 (2015).
12. Guo X, *et al.* Transfection reagent Lipofectamine triggers type I interferon signaling activation in macrophages. *Immunol Cell Biol* **97**, 92-96 (2019).
13. Gautier CA, Kitada T, Shen J. Loss of PINK1 causes mitochondrial functional defects and increased sensitivity to oxidative stress. *Proc Natl Acad Sci U S A* **105**, 11364-11369 (2008).
14. Moehlman AT, Kanfer G, Youle RJ. Loss of STING in parkin mutant flies suppresses muscle defects and mitochondria damage. *PLoS Genet* **19**, e1010828 (2023).
15. McWilliams TG, *et al.* Basal Mitophagy Occurs Independently of PINK1 in Mouse Tissues of High Metabolic Demand. *Cell Metab* **27**, 439-449.e435 (2018).
16. Sliter DA, *et al.* Parkin and PINK1 mitigate STING-induced inflammation. *Nature* **561**, 258-262 (2018).
17. Gulen MF, *et al.* cGAS-STING drives ageing-related inflammation and neurodegeneration. *Nature* **620**, 374-380 (2023).
18. McWilliams TG, *et al.* mito-QC illuminates mitophagy and mitochondrial architecture in vivo. *J Cell Biol* **214**, 333-345 (2016).
19. Dittmayer C, Völcker E, Wacker I, Schröder RR, Bachmann S. Modern field emission scanning electron microscopy provides new perspectives for imaging kidney ultrastructure. *Kidney Int* **94**, 625-631 (2018).
20. Livingston MJ, *et al.* Clearance of damaged mitochondria via mitophagy is important to the protective effect of ischemic preconditioning in kidneys. *Autophagy* **15**, 2142-2162 (2019).

21. Denic A, Glasscock RJ, Rule AD. Structural and Functional Changes With the Aging Kidney. *Adv Chronic Kidney Dis* **23**, 19-28 (2016).
22. Yeo D, Kang C, Gomez-Cabrera MC, Vina J, Ji LL. Intensified mitophagy in skeletal muscle with aging is downregulated by PGC-1alpha overexpression in vivo. *Free Radic Biol Med* **130**, 361-368 (2019).
23. Zhou J, *et al.* Changes in macroautophagy, chaperone-mediated autophagy, and mitochondrial metabolism in murine skeletal and cardiac muscle during aging. *Aging (Albany NY)* **9**, 583-599 (2017).
24. Jiménez-Loygorri JI, *et al.* Mitophagy in the retina: Viewing mitochondrial homeostasis through a new lens. *Prog Retin Eye Res* **96**, 101205 (2023).
25. Cornelissen T, Vilain S, Vints K, Gounko N, Verstreken P, Vandenberghe W. Deficiency of parkin and PINK1 impairs age-dependent mitophagy in *Drosophila*. *eLife* **7**, (2018).
26. Singh A, *et al.* Urolithin A improves muscle strength, exercise performance, and biomarkers of mitochondrial health in a randomized trial in middle-aged adults. *Cell Rep Med* **3**, 100633 (2022).
27. Andreux PA, *et al.* The mitophagy activator urolithin A is safe and induces a molecular signature of improved mitochondrial and cellular health in humans. *Nat Metab* **1**, 595-603 (2019).
28. Liu S, *et al.* Effect of Urolithin A Supplementation on Muscle Endurance and Mitochondrial Health in Older Adults: A Randomized Clinical Trial. *JAMA Netw Open* **5**, e2144279 (2022).

REVIEWER COMMENTS

Reviewer #1 (Remarks to the Author):

The revised version of the manuscript demonstrates a creditable response to the concerns raised during the review process, including the editorial ones. The authors have executed the revisions diligently, effectively addressing the points of concern that were previously identified. Furthermore, additional experiments performed, as suggested by the reviewers, have significantly enhanced the manuscript. These additional experiments have not only strengthened the paper but have also provided compelling evidence supporting the assertion that mitophagy is indeed deregulated in the aged retina. Overall, the revisions and supplementary experiments have substantially improved the quality and impact of the study, making it a valuable contribution to the field.

Reviewer #2 (Remarks to the Author):

The authors have partially addressed the previous comments, but still have some concerns. In particular, the mechanism of urolithin A remains unclear. This could be addressed in cultured cells, even if not possible in mice.

1. The addition of the lysosomal inhibition experiments using leupeptin is commendable. This suggests that the accumulation of RFP in aged mice is not due to lysosomal dysfunction. This is very important data and should be included in the paper, not just shown only to the reviewers.

2-4. It is still unclear whether the effect of urolithin A (UA) is via mitophagy. The authors now mention that urolithin A (UA) has not only mitophagy-inducing activity but also the ability to promote mitochondrial biogenesis (page 10, Fig. 6a). However, there are still many places where UA is described as a mitophagy inducer (e.g., lines 198, 202, etc.). Such biased statements are unacceptable since the authors admit that they cannot assess whether they are mitophagy-mediated or not in mice *in vivo*.

The authors have added some experiments using cultured cells to Fig. 6, but these new data contain several problems. UA treatment alone reduces the number of cytosolic DNA foci and cGAS-STING activity to control levels (Fig. 6c, d). This is surprising. It should be determined whether these effects are indeed mediated by mitophagy by knocking out or knocking down PINK1 or Parkin. It should also be determined whether the mitophagy induced by UA and ABT-737 is dependent on PINK1 or Parkin (Fig. 6e).

In Fig. 6d, it appears that pIRF3 is not activated by ABT-737 treatment. Why?

Reviewer #3 (Remarks to the Author):

The authors have largely addressed all of my previous queries with additional data, which strengthens the manuscript.

REVIEWER COMMENTS

Reviewer #1 (Remarks to the Author):

The revised version of the manuscript demonstrates a credible response to the concerns raised during the review process, including the editorial ones. The authors have executed the revisions diligently, effectively addressing the points of concern that were previously identified. Furthermore, additional experiments performed, as suggested by the reviewers, have significantly enhanced the manuscript. These additional experiments have not only strengthened the paper but have also provided compelling evidence supporting the assertion that mitophagy is indeed deregulated in the aged retina. Overall, the revisions and supplementary experiments have substantially improved the quality and impact of the study, making it a valuable contribution to the field.

We thank Reviewer #1 for their kind comments and suggestions that have significantly improved the quality and reach of our manuscript.

Reviewer #2 (Remarks to the Author):

The authors have partially addressed the previous comments, but still have some concerns. In particular, the mechanism of urolithin A remains unclear. This could be addressed in cultured cells, even if not possible in mice.

We thank Reviewer #2 for their feedback. We have now performed additional *in vivo* and *in vitro* experiments to validate our previous findings and further understand the mechanism of action of urolithin A in regards to cytosolic DNA clearance.

1. The addition of the lysosomal inhibition experiments using leupeptin is commendable. This suggests that the accumulation of RFP in aged mice is not due to lysosomal dysfunction. This is very important data and should be included in the paper, not just shown only to the reviewers.

We agree that this experiment is key and separates our results on mitophagy from previous reports on general macroautophagy and/or using alternative tandem fluorescent reporters. In the initial **Figure for Reviewers 1** we only had an n=3 for each condition, we have now treated a new cohort of Young and Old mice with Leupeptin to further validate increased mitophagy flux in the aged retina. These data are now shown as part of **Supplementary Figure 1f**.

2-4. It is still unclear whether the effect of urolithin A (UA) is via mitophagy. The authors now mention that urolithin A (UA) has not only mitophagy-inducing activity but also the ability to promote mitochondrial biogenesis (page 10, Fig. 6a). However, there are still many places where UA is described as a mitophagy inducer (e.g., lines 198, 202, etc.). Such biased statements are unacceptable since the authors admit that they cannot assess whether they are mitophagy-mediated or not in mice *in vivo*.

The authors have added some experiments using cultured cells to Fig. 6, but these new data contain several problems. UA treatment alone reduces the number of cytosolic DNA foci and cGAS-STING activity to control levels (Fig. 6c, d). This is surprising. It should be determined whether these effects are indeed mediated by mitophagy by knocking out or knocking down PINK1 or Parkin. It should also be determined whether the mitophagy induced by UA and ABT-737 is dependent on PINK1 or Parkin (Fig. 6e).

In order to address whether the effect of UA is exerted via mitophagy we first analyzed which pathway(s) were activated in ARPE-19 cells. Mirroring our findings *in vivo* (C57BL6/J mice) and *in vitro* (NHDFs), ABT-737 induced activation of PINK1/Parkin-dependent mitophagy evidenced by an increase in phospho-Ubiquitin^{Ser65} levels (**Supplementary Figure 10a**). UA also stimulated the PINK1/Parkin pathway (**Supplementary Figure 10a**) as previously reported in the literature (PMID: 33827972). Once again, we did not observe any robust alterations in the levels of proteins involved

in receptor-mediated mitophagy (**Supplementary Figure 10b**) or mitochondrial cardiolipin dynamics (**Supplementary Figure 10c**).

Taking this new data into account, we modulated PINK1/Parkin-dependent mitophagy (*PINK1*; *PARK2*-siRNA; **Figure 7f**) and/or mitochondrial biogenesis (Chloramphenicol; mitochondrial protein synthesis inhibitor; **Figure 7c**) to assess which is the main contributor to UA-associated decrease in cytosolic DNA foci. Accordingly, PINK1+Parkin knockdown abolished mitophagy stimulation by both UA and ABT-737 (**Figure 7b**). Mitochondrial biogenesis inhibition had no effect on cytosolic DNA clearance by UA in cells treated with ABT-737 (**Figure 7e**). Supporting our initial hypothesis, PINK1+Parkin knockdown fully abrogated the rescue by UA, leading to mtDNA accumulation in the cytosol after ABT-737 treatment (**Figure 7e**).

While concomitant biogenesis might be required to restore the mitochondria pool and ensure homeostasis long term, this proof-of-concept experiment validates that UA-induced PINK1/Parkin-dependent mitophagy is essential for cytosolic mtDNA clearance.

In Fig. 6d, it appears that pIRF3 is not activated by ABT-737 treatment. Why?

We apologize for the low immunoblotting quality in the previously submitted version of our manuscript. pIRF3 indeed increases in ARPE-19 cells treated with ABT-737, even though to a lesser extent than Old vs Young mice retina. We have re-run the same batch of samples increasing protein load (21 µg per lane) and decreasing primary antibody dilution (1:500), **Figure 6d** has been updated accordingly.

Reviewer #3 (Remarks to the Author):

The authors have largely addressed all of my previous queries with additional data, which strengthens the manuscript.

We thank Reviewer #3 for their kind comments and suggestions that have significantly improved the quality and reach of our manuscript.

REVIEWERS' COMMENTS

Reviewer #2 (Remarks to the Author):

The authors have adequately addressed all the points I have raised. I do not have any further comments.